# MC Layer Normalization for calibrated uncertainty in Deep Learning

**Thomas Frick**                                                                              *fri@zuirch.ibm.com*
*IBM Research, ETH Zurich*

**Diego Antognini**                                                                   *diego.antognini@ibm.com*
*IBM Research*

**Ioana Giurgiu**                                                                          *igi@zurich.ibm.com*
*IBM Research*

**Benjamin Grewe**                                                                              *bgrewe@ethz.ch*
*ETH Zurich*

**Cristiano Malossi**                                                                      *acm@zurich.ibm.com*
*IBM Research*

**Rong J.B. Zhu**                                                                      *rongzhu@fudan.edu.cn*
*Fudan University*

**Mattia Rigotti**                                                                         *mrg@zurich.ibm.com*
*IBM Research*

**Reviewed on OpenReview:** *https:// openreview. net/ forum? id= bG3ICt3EOC*

## Abstract

Efficiently estimating the uncertainty of neural network predictions has become an increasingly important challenge as machine learning models are adopted for high-stakes industrial applications where shifts in data distribution may occur. Thus, calibrated prediction uncertainty is crucial to determine when to trust a model's output and when to discard them as implausible. We propose a novel deep learning module – *MC Layer Normalization* – that acts as a drop-in replacement for Layer Normalization blocks and endows a neural network with uncertainty estimation capabilities. Our method is motivated from an approximate Bayesian perspective, but it is simple to deploy with no significant computational overhead thanks to an efficient one-shot approximation of Monte Carlo integration at prediction time. To evaluate the effectiveness of our module, we conduct experiments in two distinct settings. First, we investigate its potential to replace existing methods such as MC-Dropout and Prediction-Time Batch Normalization. Second, we explore its suitability for use cases where such conventional modules are either unsuitable or sub-optimal for certain tasks (as is the case with modules based on Batch Normalization, which is incompatible for instance with transformers). We empirically demonstrate the competitiveness of our module in terms of prediction accuracy and uncertainty calibration on established out-of-distribution image classification benchmarks, as well as its flexibility by applying it on tasks and architectures where previous methods are unsuitable. Code implementing our MC-LayerNorm module can be found here `https://github.com/IBM/mc-layernorm`.

# 1   Introduction

Endowing neural networks with a mechanism for efficient estimation of prediction uncertainty is a challenging problem that is acquiring increasing attention as these models are deployed in critical decision making settings. In high-stakes real-world applications such as autonomous driving (Bojarski et al., 2016), robotics (Sünderhauf et al., 2018), or medical diagnosis (Djuric et al., 2017; Esteva et al., 2017) models are often operating in an out-of-distribution situation compared to the training data. In such scenarios, calibrated prediction uncertainty is crucial to meaningfully compare competing predictions and decide when to trust them or when to reject them as implausible.

Bayesian methods such as Bayesian Deep Neural Networks (DNNs) offer a principled formalism to compute prediction uncertainty (MacKay, 1992). Their disadvantage is that obtaining uncertainty measures over complex models such as large neural networks quickly becomes intractable because of the computational challenge of estimating and updating posteriors over their parameters. To overcome these issues, researchers have recently proposed approximate Bayesian methods that make use of variational approximations by conveniently leveraging sampling mechanisms that are inherently present in modern DNNs such as Dropout and Batch Normalization. MC-Dropout (Gal & Ghahramani, 2016), for instance cleverly exploits the fact that Dropout noise can be interpreted as a sampling mechanism over a variational distribution approximating the posterior of the DNNs parameters given the training data. Following this work, the stochastic nature of the mini-batch sampling process exposed by Batch Normalization (BatchNorm) layers has also been used to perform approximate Bayesian inference for uncertainty estimation (Teye et al., 2018; Mukhoti et al., 2020), and domain-adaptive prediction-time calibration of uncertainty (Nado et al., 2021).

In this paper, we propose a new deep learning module that fits neatly into the line of research on sampling-based deep learning layers for estimating prediction uncertainty in neural networks. In particular, our module can be used as a drop-in replacement for Layer Normalization to seamlessly add uncertainty calibration capabilities to a neural network. Our module, named *MC Layer Normalization* (MC-LayerNorm), consists of a stochastic variant of Layer Normalization (LayerNorm) that subsamples features when computing the normalization statistics used to normalize the input features. While it offers complementary functionality to related modules such as MC-Dropout and Prediction-Time Batch Norm, it can also be used in training settings and with architectures where the latter two modules are not technically applicable or result in sub-optimal performance. In addition, MC-LayerNorm inherits the advantages of LayerNorm over BatchNorm, including the fact that its training behavior does not depend on the mini-batch size and that it is invariant to single input data re-scaling. This last property is especially appealing as it allows MC-LayerNorm to perform *zero-shot domain adaptation* at prediction time.

We illustrate how MC-LayerNorm can be theoretically motivated from an approximate Bayesian perspective while being easy to deploy in practice without imposing significant computational overhead. We then demonstrate the effectiveness of our module through empirical evaluation in two distinct settings: First, we show its competitive performance in terms of prediction accuracy and uncertainty calibration compared to state-of-the-art alternatives. To that end, we conduct experiments on established benchmarks for out-of-distribution (OOD) image classification with convolutional networks. Secondly, we highlight the applicability of MC-LayerNorm in scenarios where competing methods cannot be used due to incompatibility or suboptimality with the architectures at hand. To this end, we present experiments using the Vision Transformers on the same OOD image classification benchmarks as in the first setting. As the Vision Transformer architecture prohibits the use of Batch Norm, we show the superiority of our method compared to existing uncertainty calibration methods that can be applied. In addition, we investigate the use case of click-through rate prediction on the Criteo dataset, where the current state-of-the-art model does not include Dropout or BatchNorm layers but does include LayerNorm blocks. As a result, the use of MC-Dropout or Prediction-Time BatchNorm is impossible, while MC-LayerNorm is a natural drop-in replacement.

## 2 Related Work

Our proposed normalization layer is very much inspired by *Monte Carlo Batch Normalization* and the related work in approximate Bayesian inference in deep learning such as *MC-Dropout*, as well as the prediction-time adaptive normalization method *Prediction-Time Batch Normalization*.

*Monte Carlo Dropout* (MC-Dropout, Gal & Ghahramani (2016); Gal et al. (2017)) approximates Bayesian inference in neural networks for uncertainty estimates using dropout. Usually, Dropout sets a portion of the input features to zero at training time while not dropping any input features at test time. MC-Dropout introduces stochasticity at inference time by using the training behavior at test time. Similarly, *Monte Carlo Batch Normalization* (MCBN, Teye et al. (2018), Mukhoti et al. (2020)) introduces an alternative based on the properties of the batch normalization statistics leveraging training-time batch statistics at test time. Instead of utilizing the running batch statistics at inference time, they propose to approximate Bayesian inference by running multiple forward passes with distinct sets of training-time normalization statistics. *Prediction-Time Batch Normalization* is proposed by Nado et al. (2021) as countermeasure to the covariate shift that comes with out-of-distribution data samples. The method works by discarding the running batch statistics of batch normalization layers and instead uses the batch statistics of each separate batch at test time. Consequently, this effectively counteracts the covariate shift and significantly improves model calibration.

*Masksembles* (Durasov et al., 2021) relies on a fixed number of binary masks instead of randomly dropping parts of the network as in MC-Dropout. The masks are parameterized in a way that allows to change correlations between individual sub-models by controlling the overlap between the masks. This leads to a simple and easy to implement method with performance on par with Ensembles at a fraction of the cost. *Temperature Scaling* (Guo et al., 2017; Platt et al., 1999) is a post-hoc method which improves calibration after the initial training of the model by tuning a softmax temperature parameter on the validation/calibration set.

Recently, Rudner et al. (2023) proposed a different approach to incorporate a Bayesian viewpoint into neural network training by deriving a training objective which includes a regularization term that corresponds to performing function-space Empirical Bayes estimation. Contrary to the previous methods this approach requires to change the training objective, and therefore departs from our goal of designing a lightweight method that can be implemented as a simple drop-in replacement that otherwise leaves the training and prediction-time procedure unchanged.

A different line of related works, which also aim at efficiently estimating uncertainty of neural network predictions, but completely depart from a Bayesian viewpoint, are frequentist methods which include conformal prediction methods (Vovk et al., 2005), which have recently been successfully applied to deep learning settings (Bates et al., 2021; Zhu & Rigotti, 2021; Angelopoulos et al., 2022). They have also been developed in the direction of accommodating covariate shift, making them interesting for OOD prediction (Tibshirani et al., 2019).

## 3 MC Layer Normalization

The main contribution of this paper is to propose a new normalization module that, similar to *Monte Carlo Batch Normalization* (MCBN, Teye et al. (2018), Mukhoti et al. (2020)), also allows for uncertainty estimation, in particular through Monte Carlo sampling over a source of randomness. Crucially, instead of sampling randomness originating from the stochasticity of mini-batches such as in MCBN, we propose to use a variant of *Layer Normalization* where we inject stochasticity by subsampling features when computing the normalization statistics used to normalize the feature vector. Here we detail this procedure by first summarizing the original Layer Normalization module using a similar notation as the original paper (Ba et al., 2016).

We consider the $l$-th hidden layer in a feed-forward neural network, and let $a^l$ be the vector representation of the summed inputs to the neurons (preactivations) in that layer. These preactivations are computed through

matrix-vector multiplication of the weight matrix $W^l = (w_{ij}^l)$ and the inputs to the layer $h^l$ as:

$$a_i^l = \sum_j w_{ij}^l h_j^l, \qquad h_i^{l+1} = f(a_i^l + b_i^l), \tag{1}$$

where $f(\cdot)$ is an element-wise activation function such as ReLU and $b_i^l$ is a bias parameter.

Layer Normalization (Ba et al., 2016) was propose as a method to mitigate the covariate shift of correlated inputs as an alternative to Batch Normalization (Ioffe & Szegedy, 2015), and it consists in normalizing the hidden units in a given layer as $\bar{a}_i^l = \left(a_i^l - \mu^l\right)/\sigma^l$ using the mean $\mu^l$ and the variance $\sigma^l$ of their preactivations computed as follows:

$$\mu^l = \frac{1}{N_l} \sum_{i=1}^{N_l} a_i^l, \qquad (\sigma^l)^2 = \frac{1}{N_l} \sum_{i=1}^{N_l} \left(a_i^l - \mu^l\right)^2, \tag{2}$$

where $N_l$ is the number of units in layer $l$. One of the practical advantages of Layer Normalization over Batch Normalization is that it does not rely on any assumption about the size of the mini-batch and can therefore be used with batch size of 1.

Our MC Layer Normalization *(MC-LayerNorm)* layer consists in modifying eq. (2) by running the averages only over a random subset of units. In particular, we define a set $\mathcal{S}_l \subset [N_l] = \{1, \ldots, N_l\}$ obtained by sampling a fixed fraction $f$ of preactivations (or, equivalently, by dropping them with a drop rate $1 - f$), and compute the normalization statistics over the sampled units as follows:

$$\widetilde{\mu}^l = \frac{1}{|\mathcal{S}_l|} \sum_{i \in \mathcal{S}_l} a_i^l, \qquad (\widetilde{\sigma}^l)^2 = \frac{1}{|\mathcal{S}_l|} \sum_{i \in \mathcal{S}_l} \left(a_i^l - \widetilde{\mu}^l\right)^2, \tag{3}$$

where $|\mathcal{S}_l|$ denotes the size of the set $\mathcal{S}_l$.

As a result, the normalized preactivations computed by MC-LayerNorm and outputs of layer $l$ are:

$$\widetilde{a}_i^l = \left(a_i^l - \widetilde{\mu}^l\right)/\widetilde{\sigma}^l, \qquad \widetilde{h}_i^{l+1} = f(\widetilde{a}_i^l + b_i^l). \tag{4}$$

Notice that $\widetilde{\mu}^l$ and $\widetilde{\sigma}^l$ are random variables whose realization is determined by the subset of randomly sampled indices $\mathcal{S}_l$, and so are $\widetilde{a}_i^l$ and the outputs $\widetilde{h}_i^{l+1}$. This observation motivates the following probabilistic view of architectures endowed with *MC-LayerNorm*:

**Probabilistic view of MC-LayerNorm.**

Assuming without loss of generality a supervised learning setting on a training dataset $D = \{(x_i, y_i)\}_{i=1}^N$, we can formulate the goal of supervised learning as training parameters $\theta$ (which include the weights $W^l$ and biases $b^l$ of each layer $l$) to maximizing the likelihood of the dataset $D$ under the predictive probability

$$p_\theta(y|x) = \text{softmax}(f_\theta(x)), \tag{5}$$

where $f_\theta(\cdot)$ is parametrized as a neural network (see e.g. Gal & Ghahramani (2015)).

A network $f_\theta(x)$ endowed with MC-LayerNorm layers, can itself be modeled as a distribution of networks $f_\theta(x|\{\widetilde{\mu}^l, \widetilde{\sigma}^l\}_l)$, where $\{\widetilde{\mu}^l, \widetilde{\sigma}^l\}_l$ is the set of realizations of the random statistics in eq. (3) in all layers $l$. As we show below, an equivalent way of seeing this is to think of a sampled network $f_\theta(x|\{\widetilde{\mu}^l, \widetilde{\sigma}^l\}_l)$ for a given set $\{\widetilde{\mu}^l, \widetilde{\sigma}^l\}_l$ as a corresponding network $f_{\hat{\theta}}(x)$, where now $\hat{\theta}$ is sampled from a distribution $\hat{\theta} \sim q_\theta(\hat{\theta})$ defined appropriately. In the next section we show that the distribution $q_\theta(\hat{\theta})$ converges in a specific sense towards a Gaussian distribution around the parameters $\theta$.

**Approximate normality of MC-LayerNorm networks.**

Our main theoretical result follows directly from:

**Theorem 3.1.** *Given a network $f_\theta(x)$ with* MC-LayerNorm *layers, we denote by $\bar{f}_\theta(x)$ a corresponding network with all* MC-LayerNorm *(eq. (3)) replaced by* regular LayerNorm *(eq. (2)). Replacing LayerNorm in $\bar{f}_\theta(x)$ with* MC-LayerNorm *induces a distribution of models $f_{\hat{\theta}}(x)$ with $\hat{\theta} \sim q_\theta(\hat{\theta})$. In addition, $q_\theta(\hat{\theta})$ is asymptotically normal around the parameters $\theta$ of $\bar{f}_\theta(x)$.*

*Proof.* The proof of theorem 3.1 is sketched in appendix A.3. □

Next, we show how to use this result to approximate Bayesian inference with deep neural networks.

**Approximate Bayesian inference with MC-LayerNorm.**

The theorem 3.1 allows us to leverage the methods developed by Gal & Ghahramani (2016), who use stochastic regularization techniques to perform practical approximate inference in the space of neural networks $f_\theta(\cdot)$ by virtue of the fact that they introduce a random variable that allows for sampling over networks. Specifically, the idea starts from the goal of computing the predictive distribution $p(y|x, D)$ over outputs $y$ given a new input $x$ and the training dataset $D$ using Bayesian Model Averaging (Hoeting et al., 1999; Wilson & Izmailov, 2020), but by replacing the intractable posterior $p(\theta|D)$ over parameters with a tractable variational approximation $q^*(\theta)$, then marginalizing over $\theta$ using Monte Carlo integration:

$$p(y|x, D) = \int p_\theta(y|x)p(\theta|D)d\theta \approx \int p_\theta(y|x)q^*(\theta)d\theta \approx \frac{1}{N}\sum_{n=1}^{N} p_{\hat{\theta}_n}(y|x) \quad \text{with } \hat{\theta}_n \sim q^*(\hat{\theta}). \tag{6}$$

In practice, in a mini-batch Stochastic Grading Descent (SGD) training setting, for each training point and each MC-LayerNorm layer we sample a subset $\mathcal{S}_l$ to while computing the SGD step. This will implement the process of optimizing $q_\theta(\hat{\theta})$ towards $q^*(\hat{\theta})$.

At prediction time, what eq. (6) says is to implement Monte Carlo integration by running the network forward pass $N$ times given an input data $x$, each time with different realizations of the subsets $\mathcal{S}_l$. The obtained outputs are then averaged to obtain the final prediction for input $x$.

There is however an additional approximation of the inference process for a network with MC-LayerNorm that allows us to compute predictions even more efficiently by exploiting the second part of theorem 3.1. Because, according to the theorem the distribution $\hat{\theta} \sim q_\theta(\hat{\theta})$ is approximately normal around the parameters $\theta$ of a corresponding network with Layer Normalization, we can approximate the Monte Carlo integration in eq. (6) with:

$$\frac{1}{N}\sum_{n=1}^{N} p_{\hat{\theta}_n}(y|x) \quad \text{with } \hat{\theta}_n \sim q_\theta(\hat{\theta}) = \frac{1}{N}\sum_{n=1}^{N} \text{softmax}(f_{\hat{\theta}_n}(x)) \approx \text{softmax}(\bar{f}_\theta(x)), \tag{7}$$

where we used eq. (5) and Laplace's approximation based on the asymptotic normality of $q_\theta(\hat{\theta})$ around $\theta$ for the model $\bar{f}_\theta(x)$ with regular LayerNorm instead of MC-LayerNorm. What this suggests is that at prediction time we can simply run the model by replacing MC-LayerNorm with regular LayerNorm as a cheap further one-shot approximation of the MC integration in eq. (6). We will refer to this modality of use of MC-LayerNorm at prediction time as *One-shot approximation.*

We summarize the previous results in pseudocode snippets detailing the use of MC-LayerNorm in practice for training neural networks with SGD with backpropagation (algorithm 1), and at prediction time (algorithm 2). Notice that as for regular LayerNorm, in the case of convolutional inputs MC-LayerNorm normalizes its inputs over both channel and spatial dimensions.

**Algorithm 1** MC-LayerNorm module (training mode)

---

**Input:** input vector
$a = (a_1, \ldots, a_i, \ldots, a_N)$
**Input:** (parameter) fraction of sampled units $f$
**Sample:** Select $\mathcal{S} \subset [1, N]$ with $\lfloor f \cdot N \rfloor$ indices at random
**Compute:**
$\widetilde{\mu} = \frac{1}{|\mathcal{S}|} \sum_{i \in \mathcal{S}} a_i$
**Compute:**
$\widetilde{\sigma}^2 = \frac{1}{|\mathcal{S}|} \sum_{i \in \mathcal{S}} (a_i - \widetilde{\mu})^2$
**Output:** $\widetilde{a} = (\widetilde{a}_1, \ldots, \widetilde{a}_i, \ldots, \widetilde{a}_N)$
with $\widetilde{a}_i = \frac{a_i - \widetilde{\mu}}{\widetilde{\sigma}}$

---

**Algorithm 2** MC-LayerNorm module (eval mode)

**Input:** input vector $a = (a_1, \ldots, a_i, \ldots, a_N)$
**Input:** (optional) number of MC samples $N_c$
**if** `mc_integration` **then**
    // MC integration:
    **for** $n = 1$ **to** $N_c$ **do**
        $\widetilde{a}^n = $ `MC_LayerNorm`$(a)$     {// Run Algorithm 1}
    **end for**
    **Output:** $\bar{a} = \frac{1}{N_c} \sum_{n=1}^{N_c} \widetilde{a}^n$
**else**
    // One-shot approximation:
    **Compute:** $\mu = \frac{1}{N} \sum_{i=1}^{N} a_i$
    **Compute:** $\sigma^2 = \frac{1}{N} \sum_{i=1}^{N} (a_i - \mu)^2$
    **Output:** $\bar{a} = (\bar{a}_1, \ldots, \bar{a}_i, \ldots, \bar{a}_N)$ with $\bar{a}_i = \frac{a_i - \mu}{\sigma}$
**end if**

## 4 Results

We empirically evaluate MC-LayerNorm on a suite of established classification benchmarks. In addition to assessing the accuracy of the predictions, we verify the improved uncertainty calibration characteristics of MC-LayerNorm. We focus in particular on the out-of-distribution (OOD) setting where training and test distributions differ due to covariate shift (Shimodaira, 2000), i.e. when the marginal distributions of features are different, $p_{train}(x) \neq p_{test}(x)$, but conditional label distributions are preserved $p_{train}(y|x) = p_{test}(y|x)$. The expectation is then that properly calibrated uncertainty will be reflected in predictions that tend to be less confident on the OOD inputs that are not well represented in the training distribution.

We conduct experiments for the two distinct application settings mentioned above: Firstly, we investigate MC-LayerNorm's potential as a viable alternative to existing methods and its application to real-world scenarios with the sparse occurrence of out-of-distribution samples. Secondly, we explore MC-LayerNorm as an option for accurate uncertainty estimates in applications where competing methods cannot be used as they are incompatible with the architectures (e.g., if no BatchNorm layers are present in the model, we can't apply MC-BatchNorm).

Following previous work (Ovadia et al., 2019; Nado et al., 2021), we quantify the calibration of the uncertainty of predictions using two measures: *expected calibration error (ECE)* and *Brier score.*

**ECE**   *(lower is better)* quantifies the difference between the confidence of a model and its accuracy computed on bins of samples sorted by confidence (Naeini et al., 2015). Concretely, we group our $N$ predictions into bins $B_i$ according to confidence, compute the average prediction $\mathrm{acc}(B_i)$ within bins, then compute $\mathrm{ECE} = \sum_i \frac{B_i}{N} |\mathrm{acc}(B_i) - \mathrm{conf}(B_i)|$. As an artifact due to confidence binning, ECE can over-emphasize the tails of the probabilities (Quinonero-Candela et al., 2006), which is why the literature typically also monitors another calibration metric like the Brier score.

**Brier score**   *(lower is better)* is the squared distance between the vector of predicted probabilities and the one-hot encoded true labels (Brier, 1950). It is guaranteed to be a proper scoring rule, i.e. it decreases monotonically to zero as the predicted probabilities approach the true targets distribution (Gneiting & Raftery, 2007). Brier score plots will be presented in the Appendix.

**Image Classification**

We follow the benchmarking methodology in Nado et al. (2021): model calibration metrics (accuracy, ECE, and Brier score) are evaluated on CIFAR-10-C, TinyImageNet-C, and ImageNet-C introduced in Hendrycks & Dietterich (2019) (CCA4.0 license). Both datasets are corrupted versions of their original test sets (CIFAR-

10 (Krizhevsky, 2009), TinyImageNet (Le & Yang, 2017), and ImageNet (Deng et al., 2009)) generated by applying one of 15 corruptions (e.g., frost, motion blur, rain).

We assess model calibration for three different scenarios:

1. **In-distribution (Test):** The original test set of the corresponding dataset is used as a baseline for all metrics.

2. **Out-of-distribution (OOD):** The metrics are evaluated on the corrupted C-variant test sets (severity 5). This favors Prediction-Time BatchNorm as the batches exclusively consist of out-of-distribution samples. We use a batch size of 128 (following the insight from Nado et al. (2021) that shows that Prediction-Time BatchNorm performance degrades only for batch sizes below 128).

3. **Zero-shot prediction-time domain adaptation (Mix):** This scenario simulates the situation where a deployed model suddenly encounters OOD samples and still hasn't gathered enough observations to re-adapt (as Prediction-Time BatchNorm needs to do on multiple OOD samples). We call this "Mix" to emphasize that the models are trained on in-distribution data but tested on OOD samples. Concretely, we create batches of size $N$ consisting of $N-1$ in-distribution samples and a single out-of-distribution sample (we use the same batch size as in the OOD setting $N = 128$). The calibration metrics are then measured only on the out-of-distribution samples.

All experiments consist of fine-tuning a pretrained model with the respective uncertainty calibration method applied. We train 3 models for each hyper-parameter configuration (see Appendix A.4). Then for each normalization method we select the configuration with the best average accuracy on the validation set. No data augmentations are applied during training other than a horizontal flip of the images.

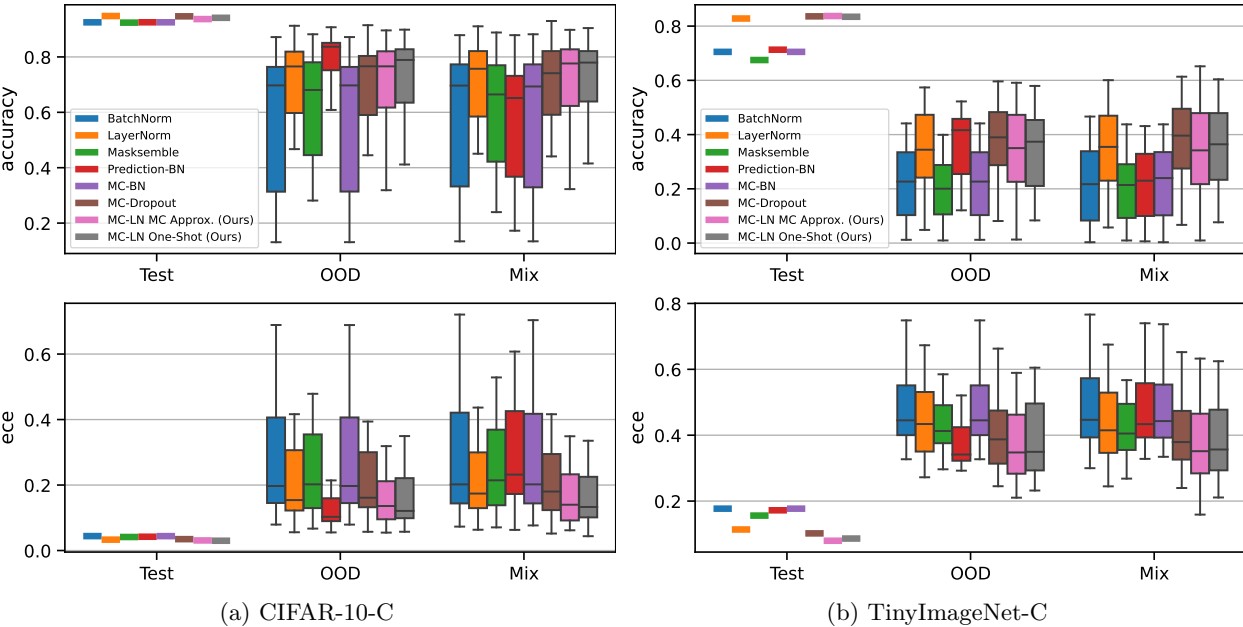

(a) CIFAR-10-C                    (b) TinyImageNet-C

Figure 1: **CIFAR-10-C and TinyImageNet-C with ConvNext: Calibration for in-distribution (Test), out-of-distribution (OOD), and zero-shot prediction-time domain adaptation (Mix) setting:** We compare LayerNorm, BatchNorm, Prediction-Time BatchNorm, MC-Dropout (f=0.9, 10 MC iterations), MC-BatchNorm (10 MC iterations), Masksembles, as well as our novel method, MC-LayerNorm (MC and One-Shot approximation, f=0.7) on the expected calibration error (ECE) and accuracy. While Prediction-Time BatchNorm is superior in the full out-of-distribution (OOD) setting, MC-LayerNorm outperforms all other methods in the zero-shot prediction-time domain adaptation (Mix) use case. Error bars are constructed from evaluation runs for the 15 corruptions applied to the original test set and over three training runs.

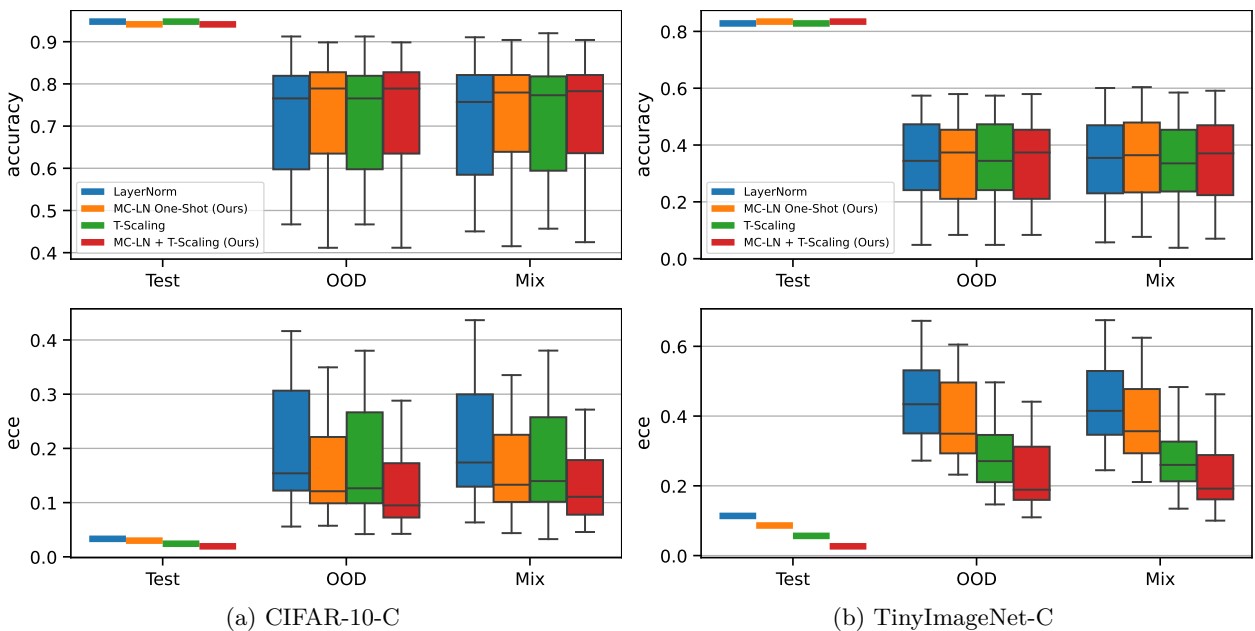

Figure 2: **Temperature Scaling ConvNext for CIFAR-10-C and TinyImageNet-C: Calibration for in-distribution (Test), out-of-distribution (OOD), and zero-shot prediction-time domain adaptation (Mix) setting:** We compare LayerNorm and our novel method, MC-LayerNorm (MC and One-Shot approximation, f=0.7) with a combination of each respective method and Temperature Scaling on the expected calibration error (ECE) and accuracy. The Temperature scaled versions outperform their respective non-post-hoc calibrated methods. Error bars are constructed from evaluation runs for the 15 corruptions applied to the original test set and over three training runs.

**Out-Of-Distribution Image Classification - ConvNext**

We evaluate the first setting by empirically comparing our MC Layer Normalization (MC integration and One-shot approximation mode) with Prediction-Time Batch Normalization introduced in Nado et al. (2021). Additionally, we report results for MC-Dropout (Gal & Ghahramani, 2016; Gal et al., 2017), MC-BatchNorm (Teye et al., 2018; Mukhoti et al., 2020), and Masksembles (Durasov et al., 2021). For both MC-Dropout and Masksembles we use the optimal parameter configuration mentioned in Durasov et al. (2021). Finally, we also investigate the combination of MC-LayerNorm with post-hoc Temperature Scaling (Guo et al., 2017; Platt et al., 1999).

We focus our evaluation on the ConvNext architecture (Liu et al., 2022), as it is the only neural network architecture for which both BatchNorm and LayerNorm normalization has been shown to reach similar image classification performance. This equality of classification accuracy leads to a fair comparison of uncertainty calibration methods independent of the used normalization layer. For CIFAR-10-C, we limit ourselves to the pre-trained architecture size "pico" from the excellent timm library (Wightman et al., 2023) (Apache-2.0 license), while for TinyImageNet-C, we make use of the "tiny" variant.

Figure 1, tables 1 and 2 shows results for CIFAR-10-C and TinyImageNet-C. While MC-LayerNorm performs slightly worse (CIFAR-10) or similarly (TinyImageNet) in terms of calibration (ECE) compared to Prediction-Time BatchNorm in the OOD setting, it outperforms all other methods in the zero-shot prediction-time domain adaptation (Mix) setting. Prediction-Time BatchNorm loses much of its real-world (Mix) performance because it relies heavily on batches consisting entirely of out-of-distribution samples. In contrast, LayerNorm methods do not rely on batch statistics as they calculate normalization statistics across channels and spatial dimensions. As a result, our method's calibration abilities are independent of batch size and work for small but also for mixed in- and out-of-distribution batches.

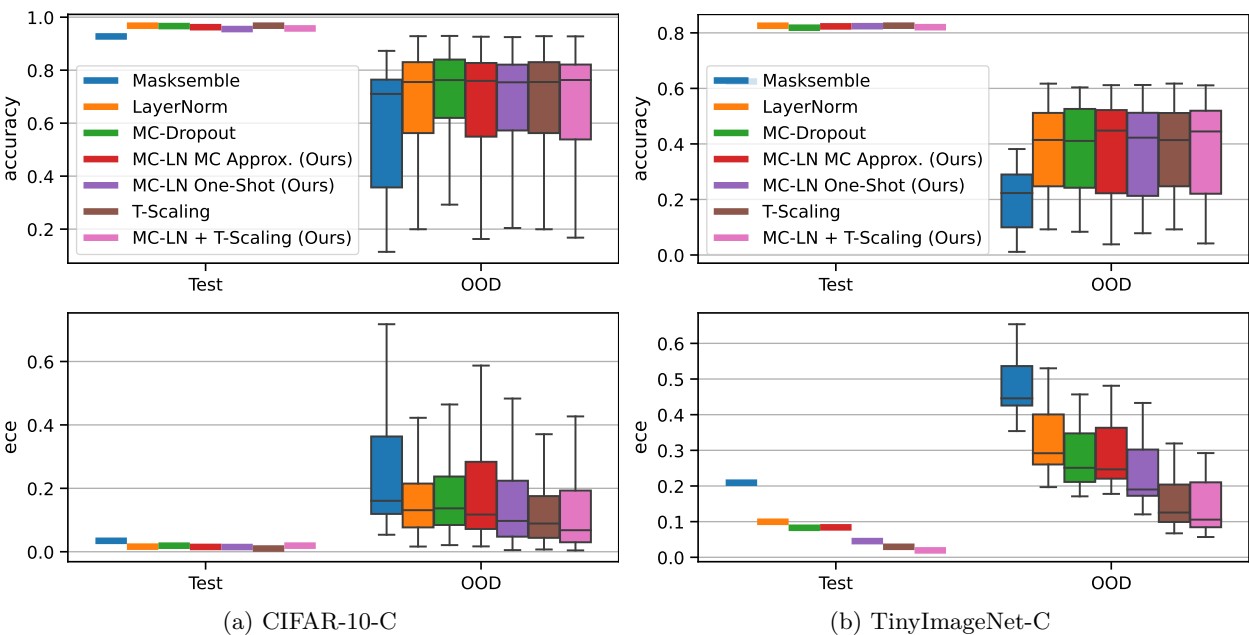

(a) CIFAR-10-C

(b) TinyImageNet-C

Figure 3: **CIFAR-10-C and TinyImageNet-C with Vision Transformer: Calibration for in-distribution (Test) and out-of-distribution (OOD):** We compare LayerNorm, MC-Dropout (f=0.9, 10 MC iterations), MC-BatchNorm (10 MC iterations), Masksembles, Temperature Scaling as well as our novel method - MC-LayerNorm (MC and One-Shot approximation, f=0.6) - on the expected calibration error (ECE) and accuracy. MC-LayerNorm in combination with Temperature scaling outperforms all other methods. Error bars are constructed from evaluation runs for the 15 corruptions applied to the original test set and over three training runs.

Figure 5 and 6 illustrate calibration performance comparisons for varying fractions of sampled features of the MC-LayerNorm. The results show that MC-LayerNorm is stable with respect to its only hyper-parameter, the fraction of subsampled features. The model performance and calibration remain stable even for configurations where the normalization is calculated from only 60% of the input features.

Next, we evaluate the necessity of enhancing approximate Bayesian methods with post-hoc calibration. Figure 2 shows the calibration performance of LayerNorm and MC-LayerNorm as well as the combination of these methods with post-hoc Temperature Scaling. We observe that the Temperature scaled versions surpass their respective post-hoc calibrated counterparts. Temperature scaling alone shows a strong effect as it equals (CIFAR-10) or outperforms (TinyImageNet) MC-LayerNorm for both the OOD and Mix setting. However, the combined effect of Temperature Scaling and MC-LayerNorm is noticeably more robust and results in a better overall calibration, indicating practically interesting synergistic effects between the two methods. This pattern is also evident in experiments with Vision Transformers, as shown in Figure 3.

We see approximate Bayesian estimation methods (e.g., our MC-LayerNorm) and post-hoc Temperature Scaling as complementary and synergistic. For one, post-hoc calibration is not always possible, as it requires a held-out in-distribution calibration dataset, which might not always be available. In such a case, one must necessarily rely on approaches like approximate Bayesian estimation methods. Second, post-hoc calibration only works if the base model is already at least partially calibrated to the extent that the miscalibration pattern is consistent across the range of logits. The better the base model is calibrated, the more effective post-hoc calibration methods will be. Here, we believe that approximate Bayesian models can certainly be crucial to enable further calibration through post-hoc methods.

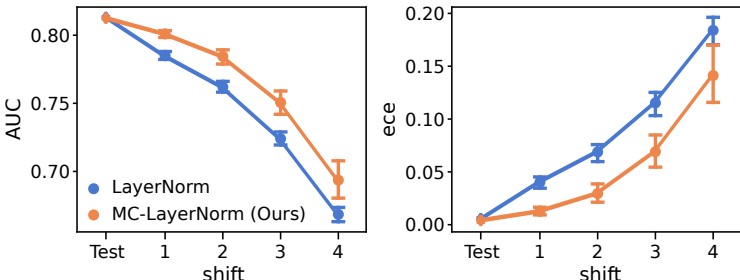

Figure 4: **Criteo dataset experiment. Area under the Curve (AUC) and expected calibration error (ECE) for four shift intensities:** vanilla LayerNorm is compared to MC-LayerNorm (f=0.7) while applying increasing amounts noise to the test set. MC-LayerNorm shows a clear advantage, retaining more of the original model performance as the data shifts further out-of-distribution (plots show average over 10 runs; error bars indicate standard errors around the mean).

### Out-Of-Distribution Image Classification - Vision Transformer

We evaluate setting 2 by running experiments using the Vision Transformer (Dosovitskiy et al., 2021) architecture (ViT). As discussed in more detail in Shen et al. (2020), batch norm is not used for ViTs for a variety of historical and performance reasons. Therefore, endowing them with MC-BatchNorm or Prediction-Time BatchNorm would require a relatively radical architectural alteration which will typically enforce a change in the training procedure (specifically, to take into account the effect of mini-batch size on the training with BatchNorm), and compromise on the performance as state-of-the-art is empirically achieved without a BatchNorm layer. Thus MC-LayerNorm is the only norm layer based uncertainty calibration method that is applicable to this architecture. We use the pre-trained architecture implementations from timm (Wightman et al., 2023) (Apache-2.0 license) and use the size "vit tiny" and "vit small" for CIFAR10 and TinyImageNet/ImageNet respectively. We compare our method's performance (in MC integration and One-shot approximation mode) versus MC-Dropout (Gal & Ghahramani, 2016; Gal et al., 2017), Temperature Scaling (Guo et al., 2017; Platt et al., 1999) and Masksembles (Durasov et al., 2021). For both MC-Dropout and Masksembles we use the optimal parameter configuration mentioned in Durasov et al. (2021). Additionally, we also report results for a combination of Temperature Scaling and our MC-LayerNorm.

Figure 3, tables 3 and table 4 shows the superiority of our method when evaluated in terms of uncertainty calibration and accuracy for the Test and the OOD setting. As non of the evaluated methods are batch dependent, unlike the BatchNorm methods from the previous experiment, we forgo evaluation on the zero-shot prediction-time domain adaptation use case in this section. MC-LayerNorm outperforms classical Layer-Norm, MC-Dropout and Masksembles in terms of ECE and accuracy on both setting. While Temperature Scaling outperforms MC-LayerNorm, we can show that a combination of MC-LayerNorm and post-hoc Temperature scaling reaches maximum calibration performance. These results are consistent with the additional experiments in Appendix A.2.1.

Training ViTs with the Masksemble approach was unfortunately difficult as non of the models we trained converged to a reasonable classification performance. We hypothesize that ViT token width is too narrow for this method to converge as it splits the token into N subsets with S overlap.

Figure 7a and 7b shows calibration performance comparisons for varying fractions of sampled features of the MC-LayerNorm.

### Criteo Display Advertising Challenge

Here, we showcase the value of our method in another situation where the competing methods cannot be used as they are incompatible with the architectures. For the experiments, we train models on the Criteo Display Advertising Challenge (CriteoLabs, 2014). We leverage the current state-of-the-art model, MaskNet (Wang et al., 2021), relying on the implementation from (Zhu (2023), Apache-2.0 license). This

architecture does not include any BatchNorm layers, meaning that it cannot be naturally extended with a MC-BatchNorm/Predition-Time BatchNorm without modifying the training procedure (specifically, to take into account the effect of mini-batch size on the training with BatchNorm), and without incurring a loss in performance (as SOTA is empirically achieved without a BatchNorm layer). MaskNet however includes LayerNorm modules, meaning that we can straightforwardly replace them by dropping in MC-LayerNorm without changing anything else in the architecture or training. The Criteo challenge is therefore also meant as a use case to emphasize the advantages of our approach also beyond vision tasks, in particular its flexibility as a drop-in replacement layer for LayerNorm (a normalization layer which is almost ubiquitous in modern deep learning) and its ease of use since it does not require any other modifications to the models architecture. We train two types of models: the original MaskNet model containing vanilla LayerNorm and a patched model for which we replace all LayerNorm layers with MC-LayerNorm. At prediction time, we run the MC-LayerNorm models using the *One-shot approximation*, making them computationally as cheap as the regular MaskNet models to evaluate. The trained models are evaluated on the original test set and a corrupted version of the test set. We simulate out-of-distribution data by applying corruption in the form of Gaussian noise on the numerical features. Shift intensities $[6.25\%, 12.5\%, 25\%, 50\%]$ are based on units of the standard deviation of the original feature values. Finally, we measure the difference in model performance between the original models with LayerNorm and the adjusted models with MC-LayerNorm.

Figure 4 shows results on the Criteo dataset demonstrating a clear advantage of MC-LayerNorm when it come the considered metrics. Here we displays the performance of MC-LayerNorm for f=0.7 (the curves are robust within the interval f=0.6-0.8, with a gradual progressive degradation as f is varied above or below this range). The Figure shows that as we shift the tests set further out-of-distribution, our method retains more of the original calibration and model performance compared to the baseline of the traditional LayerNorm.

## 5 Conclusion and Discussion

We presented a novel normalization layer, MC-LayerNorm, which improves model calibration for in- and out-of-distribution (OOD) data by injecting stochasticity through subsampling features when computing the normalization statistics. Our method can be easily deployed without significant computational overhead by replacing the LayerNorm blocks in an already trained model and then fine-tuning it. We empirically demonstrated the application of MC-LayerNorm for two distinct settings: First, we have shown its potential as an alternative to existing methods for OOD classification and its superiority compared to Batch Norm based alternatives when it comes to mixed in- and out-of-distribution batches. Secondly, we presented experiments for use cases where application of conventional methods are either unsuitable or technically infeasible. In this setting, MC-LayerNorm was shown to outperform alternatives on OOD classification using Vision Transformers and on the CTR Criteo dataset using MaskNet.

In other words, MC-LayerNorm is able to perform zero-shot prediction-time domain adaptation, which gives it an advantage in real-world OOD use cases where a deployed model is being exposed for the first time to new OOD samples and has not had time to properly adapt to the covariate shift.

**Limitations and Future Work.** In addition to the C-variants of the considered datasets we focused on in this study, future work could validate our method on their respective A-variants, which consist of real-world adversarial examples. An additional direction for future work is MC-LayerNorm's potential impact on other model performance metrics, such as the stability of training convergence. For example, during experimentation, we noticed improved training convergence toward the same performance metrics (reduced variability across runs with different hyper-parameter settings like different learning rate values) when employing MC-LayerNorm.

In terms of broader impact, as already discussed, being able to provide deep neural networks with calibrated prediction uncertainty would make their deployment potentially safer and trusted, particularly in high-stakes applications where it is crucial to be able to reliably take consequential decisions based on them. Therefore, we argue that the line of research to which our work contributes is an important step towards trying to systematically mitigate potential negative societal effects due to the involuntary misuse of deep learning technologies.

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

# A  Appendix

## A.1  Additional results

### A.1.1  Comparison of MC-LayerNorm drop rates for ConvNext

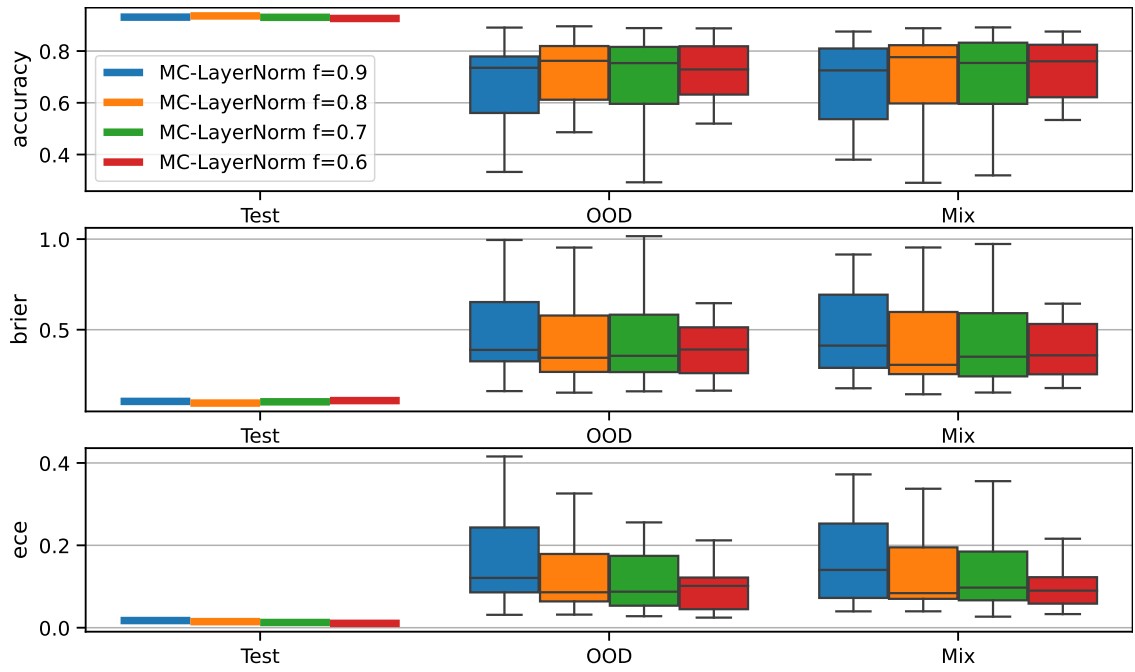

Figure 5: **CIFAR-10-C with ConvNext:** Comparison of calibration between different MC-LayerNorm (One-Shot approximation) subsampling fractions for in-distribution (Test), out-of-distribution (OOD) and the zero-shot prediction-time domain adaptation (Mix) setting.

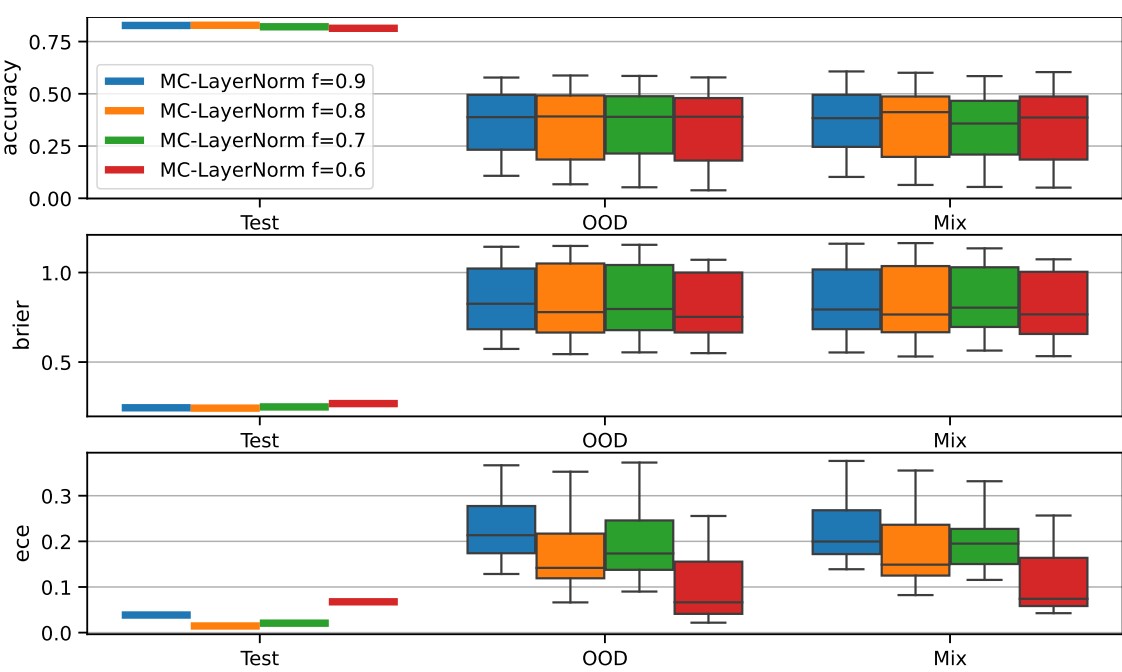

Figure 6: **TinyImageNet-C with ConvNext**: Comparison of calibration between different MC-LayerNorm (One-Shot approximation) subsampling fractions for in-distribution (Test), out-of-distribution (OOD) and the zero-shot prediction-time domain adaptation (Mix) setting.

## A.2 Comparison of MC-LayerNorm drop rates for Vision Transformers

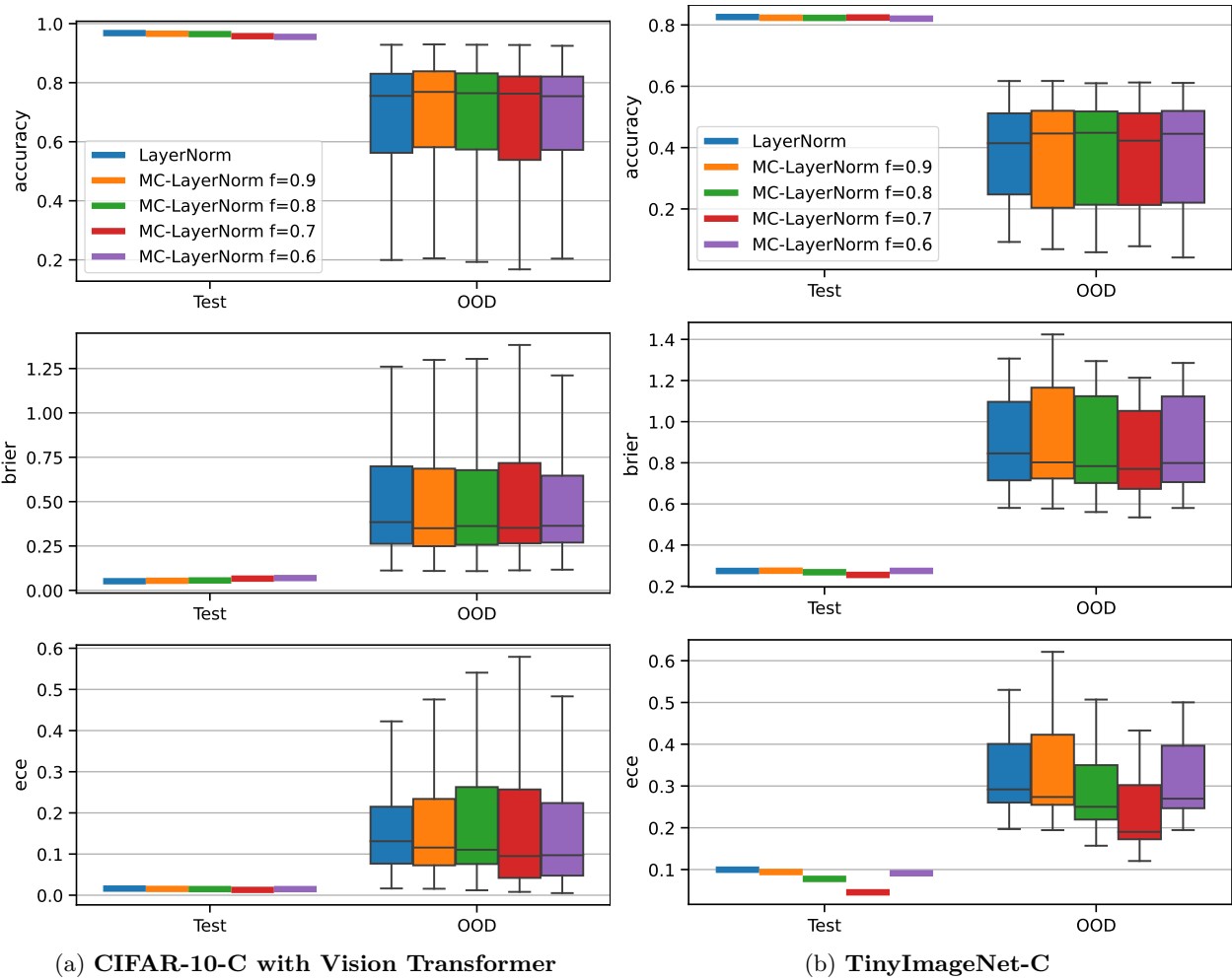

(a) **CIFAR-10-C with Vision Transformer**

(b) **TinyImageNet-C**

Figure 7: Comparison of calibration between different MC-LayerNorm (One-Shot approximation) subsampling fractions for in-distribution (Test) and out-of-distribution (OOD).

### A.2.1 ImageNet Results for Vision Transformers

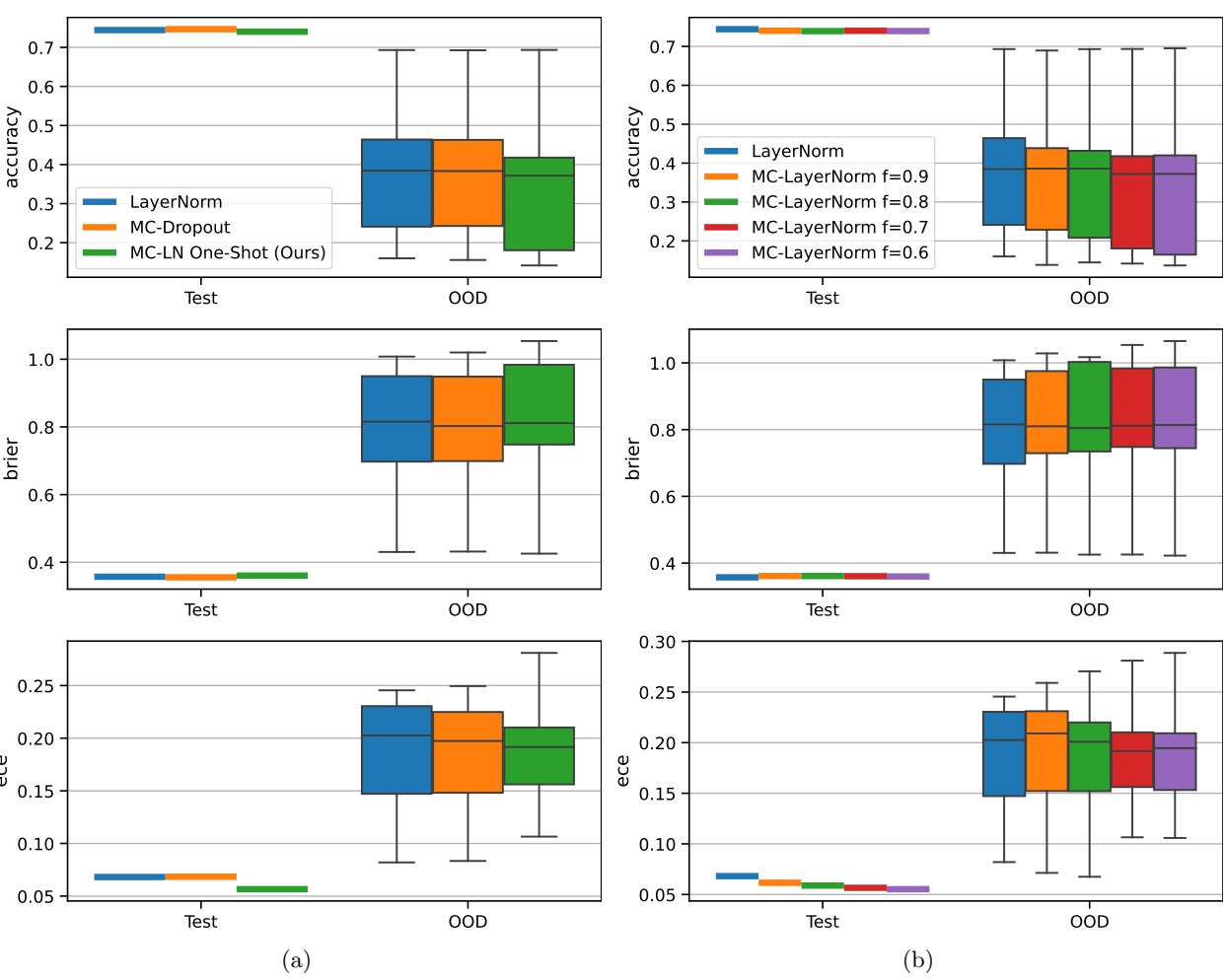

(a)  (b)

Figure 8: **ImageNet-C with Vision Transformers:**: (a) We compare LayerNorm, MC-Dropout (f=0.9, 10 MC iterations), and our novel method - MC-LayerNorm (f=0.7, One-Shot approximation) - on the expected calibration error (ECE), brier score, and accuracy. We do not include results for Masksembles in this figure due to bad performance. Error bars are constructed from evaluation runs for the 15 corruptions applied to the original test set and over three training runs. (b) Comparison of calibration between different MC-LayerNorm (One-Shot approximation) subsampling fractions for in-distribution (Test) and out-of-distribution (OOD).

### A.2.2 Result Tabels

| Scenario | Method | ECE ↓ | Brier ↓ | Accuracy ↑ |
|---|---|---|---|---|
| Test | BatchNorm | $0.044 \pm 0.001$ | $0.120 \pm 0.001$ | $0.925 \pm 0.001$ |
| | LayerNorm | $0.033 \pm 0.002$ | $0.087 \pm 0.004$ | $\mathbf{0.948 \pm 0.001}$ |
| | Masksemble | $0.041 \pm 0.001$ | $0.120 \pm 0.001$ | $0.924 \pm 0.001$ |
| | Prediction-BN | $0.042 \pm 0.002$ | $0.119 \pm 0.001$ | $0.925 \pm 0.002$ |
| | MC-BN | $0.044 \pm 0.001$ | $0.120 \pm 0.001$ | $0.925 \pm 0.001$ |
| | MC-Dropout | $0.035 \pm 0.001$ | $0.089 \pm 0.002$ | $0.947 \pm 0.001$ |
| | T-Scaling | $0.024 \pm 0.003$ | $\mathbf{0.083 \pm 0.003}$ | $\mathbf{0.948 \pm 0.001}$ |
| | MC-LN MC Approx (Ours) | $0.031 \pm 0.001$ | $0.098 \pm 0.002$ | $0.937 \pm 0.002$ |
| | MC-LN One-Shot (Ours) | $0.030 \pm 0.002$ | $0.093 \pm 0.003$ | $0.941 \pm 0.003$ |
| | MC-LN + T-Scaling (Ours) | $\mathbf{0.019 \pm 0.002}$ | $0.090 \pm 0.003$ | $0.941 \pm 0.003$ |
| OOD | BatchNorm | $0.276 \pm 0.030$ | $0.655 \pm 0.043$ | $0.592 \pm 0.014$ |
| | LayerNorm | $0.206 \pm 0.012$ | $0.479 \pm 0.026$ | $0.712 \pm 0.015$ |
| | Masksemble | $0.237 \pm 0.009$ | $0.587 \pm 0.014$ | $0.627 \pm 0.007$ |
| | Prediction-BN | $\mathbf{0.122 \pm 0.003}$ | $\mathbf{0.308 \pm 0.008}$ | $\mathbf{0.809 \pm 0.005}$ |
| | MC-BN | $0.276 \pm 0.030$ | $0.655 \pm 0.043$ | $0.592 \pm 0.014$ |
| | MC-Dropout | $0.200 \pm 0.004$ | $0.468 \pm 0.007$ | $0.716 \pm 0.005$ |
| | T-Scaling | $0.174 \pm 0.012$ | $0.454 \pm 0.025$ | $0.712 \pm 0.015$ |
| | MC-LN MC Approx (Ours) | $0.166 \pm 0.013$ | $0.454 \pm 0.033$ | $0.703 \pm 0.023$ |
| | MC-LN One-Shot (Ours) | $0.158 \pm 0.014$ | $0.419 \pm 0.032$ | $0.731 \pm 0.022$ |
| | MC-LN + T-Scaling (Ours) | $0.125 \pm 0.012$ | $0.399 \pm 0.030$ | $0.731 \pm 0.022$ |
| Mix | BatchNorm | $0.277 \pm 0.032$ | $0.651 \pm 0.046$ | $0.592 \pm 0.017$ |
| | LayerNorm | $0.212 \pm 0.013$ | $0.484 \pm 0.029$ | $0.709 \pm 0.018$ |
| | Masksemble | $0.253 \pm 0.024$ | $0.610 \pm 0.035$ | $0.614 \pm 0.012$ |
| | Prediction-BN | $0.288 \pm 0.027$ | $0.681 \pm 0.048$ | $0.576 \pm 0.024$ |
| | MC-BN | $0.276 \pm 0.025$ | $0.649 \pm 0.036$ | $0.596 \pm 0.012$ |
| | MC-Dropout | $0.202 \pm 0.004$ | $0.467 \pm 0.007$ | $0.718 \pm 0.007$ |
| | T-Scaling | $0.173 \pm 0.009$ | $0.446 \pm 0.023$ | $0.718 \pm 0.013$ |
| | MC-LN MC Approx (Ours) | $0.166 \pm 0.009$ | $0.445 \pm 0.026$ | $0.708 \pm 0.021$ |
| | MC-LN One-Shot (Ours) | $0.159 \pm 0.015$ | $0.414 \pm 0.035$ | $\mathbf{0.734 \pm 0.023}$ |
| | MC-LN + T-Scaling (Ours) | $\mathbf{0.133 \pm 0.009}$ | $\mathbf{0.400 \pm 0.030}$ | $\mathbf{0.734 \pm 0.022}$ |

Table 1: CIFAR-10-C with ConvNext: Comparison of calibration and accuracy for in-distribution (Test), out-of-distribution (OOD) and the zero-shot prediction-time domain adaptation (Mix) setting. In contrast to the figures, the standard deviation is computed distinctively by first averaging across multiple corruption levels, followed by calculating the standard deviation across various trained models.

| Scenario | Method | ECE ↓ | Brier ↓ | Accuracy ↑ |
|---|---|---|---|---|
| Test | BatchNorm | $0.177 \pm 0.006$ | $0.461 \pm 0.015$ | $0.705 \pm 0.007$ |
| | LayerNorm | $0.114 \pm 0.003$ | $0.280 \pm 0.006$ | $0.828 \pm 0.003$ |
| | Masksemble | $0.156 \pm 0.004$ | $0.478 \pm 0.003$ | $0.675 \pm 0.002$ |
| | Prediction-BN | $0.172 \pm 0.002$ | $0.450 \pm 0.001$ | $0.713 \pm 0.002$ |
| | MC-BN | $0.177 \pm 0.006$ | $0.461 \pm 0.015$ | $0.705 \pm 0.007$ |
| | MC-Dropout | $0.102 \pm 0.001$ | $0.262 \pm 0.001$ | $0.836 \pm 0.002$ |
| | T-Scaling | $0.056 \pm 0.001$ | $0.252 \pm 0.006$ | $0.828 \pm 0.003$ |
| | MC-LN MC Approx (Ours) | $0.080 \pm 0.001$ | $0.245 \pm 0.002$ | $\mathbf{0.837 \pm 0.003}$ |
| | MC-LN One-Shot (Ours) | $0.086 \pm 0.003$ | $0.252 \pm 0.003$ | $0.834 \pm 0.003$ |
| | MC-LN + T-Scaling (Ours) | $\mathbf{0.027 \pm 0.001}$ | $\mathbf{0.234 \pm 0.001}$ | $0.834 \pm 0.003$ |
| OOD | BatchNorm | $0.478 \pm 0.013$ | $1.177 \pm 0.016$ | $0.219 \pm 0.005$ |
| | LayerNorm | $0.443 \pm 0.020$ | $1.049 \pm 0.032$ | $0.337 \pm 0.013$ |
| | Masksemble | $0.426 \pm 0.004$ | $1.144 \pm 0.004$ | $0.195 \pm 0.002$ |
| | Prediction-BN | $0.373 \pm 0.001$ | $0.956 \pm 0.002$ | $0.362 \pm 0.004$ |
| | MC-BN | $0.478 \pm 0.013$ | $1.177 \pm 0.016$ | $0.219 \pm 0.005$ |
| | MC-Dropout | $0.403 \pm 0.015$ | $0.982 \pm 0.021$ | $\mathbf{0.367 \pm 0.006}$ |
| | T-Scaling | $0.276 \pm 0.021$ | $0.904 \pm 0.027$ | $0.337 \pm 0.013$ |
| | MC-LN MC Approx (Ours) | $0.369 \pm 0.001$ | $0.970 \pm 0.002$ | $0.339 \pm 0.003$ |
| | MC-LN One-Shot (Ours) | $0.381 \pm 0.004$ | $0.980 \pm 0.002$ | $0.345 \pm 0.003$ |
| | MC-LN + T-Scaling (Ours) | $\mathbf{0.226 \pm 0.005}$ | $\mathbf{0.861 \pm 0.003}$ | $0.345 \pm 0.003$ |
| Mix | BatchNorm | $0.480 \pm 0.014$ | $1.182 \pm 0.016$ | $0.219 \pm 0.003$ |
| | LayerNorm | $0.437 \pm 0.023$ | $1.039 \pm 0.037$ | $0.340 \pm 0.016$ |
| | Masksemble | $0.420 \pm 0.018$ | $1.133 \pm 0.026$ | $0.200 \pm 0.008$ |
| | Prediction-BN | $0.472 \pm 0.005$ | $1.175 \pm 0.009$ | $0.215 \pm 0.001$ |
| | MC-BN | $0.478 \pm 0.019$ | $1.175 \pm 0.025$ | $0.221 \pm 0.009$ |
| | MC-Dropout | $0.404 \pm 0.016$ | $0.985 \pm 0.020$ | $\mathbf{0.367 \pm 0.005}$ |
| | T-Scaling | $0.278 \pm 0.023$ | $0.910 \pm 0.032$ | $0.335 \pm 0.014$ |
| | MC-LN MC Approx (Ours) | $0.370 \pm 0.004$ | $0.970 \pm 0.006$ | $0.339 \pm 0.007$ |
| | MC-LN One-Shot (Ours) | $0.381 \pm 0.008$ | $0.984 \pm 0.008$ | $0.343 \pm 0.008$ |
| | MC-LN + T-Scaling (Ours) | $\mathbf{0.229 \pm 0.005}$ | $\mathbf{0.857 \pm 0.008}$ | $0.345 \pm 0.009$ |

Table 2: Tiny-ImageNet with ConvNext: Comparison of calibration and accuracy for in-distribution (Test), out-of-distribution (OOD) and the zero-shot prediction-time domain adaptation (Mix) setting. In contrast to the figures, the standard deviation is computed distinctively by first averaging across multiple corruption levels, followed by calculating the standard deviation across various trained models.

| Scenario | Method | ECE ↓ | Brier ↓ | Accuracy ↑ |
|---|---|---|---|---|
| Test | LayerNorm | $0.016 \pm 0.003$ | $0.051 \pm 0.003$ | $\mathbf{0.968 \pm 0.003}$ |
| | MC-Dropout | $0.019 \pm 0.003$ | $0.054 \pm 0.001$ | $0.966 \pm 0.001$ |
| | Masksemble | $0.034 \pm 0.003$ | $0.114 \pm 0.006$ | $0.927 \pm 0.005$ |
| | T-Scaling | $\mathbf{0.010 \pm 0.000}$ | $\mathbf{0.050 \pm 0.004}$ | $\mathbf{0.968 \pm 0.003}$ |
| | MC-LN Approx (Ours) | $0.015 \pm 0.004$ | $0.060 \pm 0.005$ | $0.962 \pm 0.004$ |
| | MC-LN One-Shot (Ours) | $0.015 \pm 0.003$ | $0.069 \pm 0.013$ | $0.955 \pm 0.010$ |
| | MC-LN + T-Scaling (Ours) | $0.019 \pm 0.015$ | $0.067 \pm 0.016$ | $0.958 \pm 0.010$ |
| OOD | LayerNorm | $0.194 \pm 0.020$ | $0.490 \pm 0.025$ | $\mathbf{0.690 \pm 0.011}$ |
| | MC-Dropout | $0.206 \pm 0.020$ | $0.499 \pm 0.024$ | $0.688 \pm 0.006$ |
| | Masksemble | $0.259 \pm 0.020$ | $0.621 \pm 0.029$ | $0.614 \pm 0.011$ |
| | T-Scaling | $0.154 \pm 0.022$ | $\mathbf{0.461 \pm 0.022}$ | $\mathbf{0.690 \pm 0.011}$ |
| | MC-LN Approx (Ours) | $0.211 \pm 0.023$ | $0.522 \pm 0.037$ | $0.673 \pm 0.020$ |
| | MC-LN One-Shot (Ours) | $0.180 \pm 0.018$ | $0.493 \pm 0.015$ | $0.679 \pm 0.018$ |
| | MC-LN + T-Scaling (Ours) | $\mathbf{0.146 \pm 0.024}$ | $0.469 \pm 0.011$ | $0.686 \pm 0.016$ |

Table 3: CIFAR-10 for Vision Transformers: Comparison of calibration and accuracy for in-distribution (Test), out-of-distribution (OOD) and the zero-shot prediction-time domain adaptation (Mix) setting. In contrast to the figures, the standard deviation is computed distinctively by first averaging across multiple corruption levels, followed by calculating the standard deviation across various trained models.

| Scenario | Method | ECE ↓ | Brier ↓ | Accuracy ↑ |
|---|---|---|---|---|
| Test | LayerNorm | $0.100 \pm 0.003$ | $0.274 \pm 0.006$ | $\mathbf{0.826 \pm 0.004}$ |
| | MC-Dropout | $0.083 \pm 0.002$ | $0.276 \pm 0.003$ | $0.819 \pm 0.003$ |
| | Masksemble | $0.209 \pm 0.003$ | $0.568 \pm 0.004$ | $0.625 \pm 0.002$ |
| | T-Scaling | $0.029 \pm 0.000$ | $\mathbf{0.253 \pm 0.005}$ | $\mathbf{0.826 \pm 0.004}$ |
| | MC-LN Approx. (Ours) | $0.084 \pm 0.002$ | $0.268 \pm 0.002$ | $0.823 \pm 0.003$ |
| | MC-LN One-Shot (Ours) | $0.045 \pm 0.003$ | $0.255 \pm 0.005$ | $0.824 \pm 0.005$ |
| | MC-LN + T-Scaling (Ours) | $\mathbf{0.019 \pm 0.000}$ | $0.256 \pm 0.003$ | $0.820 \pm 0.001$ |
| OOD | LayerNorm | $0.325 \pm 0.010$ | $0.905 \pm 0.017$ | $0.377 \pm 0.012$ |
| | MC-Dropout | $0.272 \pm 0.005$ | $0.864 \pm 0.004$ | $0.375 \pm 0.003$ |
| | Masksemble | $0.483 \pm 0.008$ | $1.201 \pm 0.010$ | $0.197 \pm 0.002$ |
| | T-Scaling | $0.152 \pm 0.008$ | $0.796 \pm 0.014$ | $0.377 \pm 0.011$ |
| | MC-LN Approx. (Ours) | $0.289 \pm 0.007$ | $0.875 \pm 0.005$ | $0.373 \pm 0.006$ |
| | MC-LN One-Shot (Ours) | $0.226 \pm 0.011$ | $0.834 \pm 0.015$ | $0.372 \pm 0.011$ |
| | MC-LN + T-Scaling (Ours) | $\mathbf{0.141 \pm 0.005}$ | $\mathbf{0.792 \pm 0.005}$ | $\mathbf{0.372 \pm 0.005}$ |

Table 4: Tiny-ImageNet for Vision Transformers: Comparison of calibration and accuracy for in-distribution (Test), out-of-distribution (OOD) and the zero-shot prediction-time domain adaptation (Mix) setting. In contrast to the figures, the standard deviation is computed distinctively by first averaging across multiple corruption levels, followed by calculating the standard deviation across various trained models.

### A.3 Proof of theorem 3.1

**Lemma A.1.** *Denote $V_\mu = \frac{1}{N_l-1}\sum_{i=1}^{N_l}(a_i^l - \mu^l)^2$ and $V_{\sigma^2} = \frac{1}{N_l-1}\sum_{i=1}^{N_l}\left((a_i^l)^2 - (\mu^l)^2 - \sigma_l^2\right)^2$. Assume $N_l - |\mathcal{S}_l| \to \infty$ as $|\mathcal{S}_l| \to \infty$.*

*(1) Assume that $\frac{1}{N_l}\sum_{i=1}^{N_l}(a_i^l)^2$ is bounded, then we have that, as $|\mathcal{S}_l| \to \infty$,*

$$|\mathcal{S}_l|^{1/2}(\widetilde{\mu}^l - \mu^l) \to_d \mathcal{N}(0, V_\mu). \tag{8}$$

*(2) Assume that $\frac{1}{N_l}\sum_{i=1}^{N_l}(a_i^l)^4$ is bounded, then we have that, as $|\mathcal{S}_l| \to \infty$,*

$$|\mathcal{S}_l|^{1/2}\left((\widetilde{\sigma}^l)^2 - (\sigma^l)^2\right) \to_d \mathcal{N}(0, V_{\sigma^2}). \tag{9}$$

*Proof.* The proof follows from Hájek (1960) under standard regularity conditions. □

**Proposition A.2.** *Under the assumption that $f(\cdot)$ in eq. (1) and eq. (4) is differentiable almost everywhere, theorem A.1 implies that there exists a variance $V$ such that*

$$|\mathcal{S}_l|^{1/2}\left(\widetilde{h}_i^{l+1} - h_i^{l+1}\right) \to_d \mathcal{N}(0, V). \tag{10}$$

*Proof.* The proof follows from the delta method in Vaart (1998). □

Proposition A.2 means that the effect of MC Layer Normalization is to approximately add Gaussian noise to the activations of a corresponding model with Layer Normalization.

### A.4 Hyperparameters

All models are trained with the AdamW optimizer provided by the timm library Wightman et al. (2023) using default parameters for $\beta_1$, $\beta_2$ and $\epsilon$. We use grid-search to tune the learning rate from values between $[1e-3, 1e-5]$ and for weight decay form values between $[0.1, 1e-4]$.

For CIFAR-10 and TinyImageNet, we run fine-tuning for 20 epochs with a batch size of 512 and with a constant learning rate. Meanwhile, for ImageNet, we run fine-tuning for 5 epochs with a batch size of 256 and constant learning rate. For Masksemble models we extend learning to 20 epochs as the needed changes to the network are much bigger than for the other methods. Finally, we run training for 100 epochs with a batch size of 1000 reducing the learning rate by 0.1 on a plateau with patience 2 checked on the validation set for Criteo.

All vision models are trained with absolutely basic data augmentation: We use padding + random crop for CIFAR-10 and TinyImageNet while only taking a random crop for ImageNet. Additionally, we use a horizontal flip (p=0.5) augmentation for all vision datasets. No training augmentations are used for the Criteo models.

For Temperature scaling (Guo et al., 2017) we use parameters proposed by Pleiss (2024): NLL Loss, LBFGS as an Optimizer with learning rate of 0.01 and running for a maximum of 50 iterations. For each dataset, a fixed set of samples from the training set is set aside as a calibration set.

All models were trained on an internal cluster consisting of NVIDIA A100 GPUs.

