# OpenReview forum: "MC Layer Normalization for calibrated uncertainty in Deep Learning"
_TMLR — Accepted by TMLR_

### Review · Reviewer_fKNX · 2023-12-18

**Summary Of Contributions:**

This manuscript develops a new method for calibrating neural network predictions via uncertainty estimation. The proposed approach builds upon Layer Normalization, introducing stochasticity into the computation of the normalization statistics by subsampling the preactivation features. This stochasticity is then exploited to obtain a Monte Carlo estimate of the predictive distribution (hence the name MC-LayerNorm). In addition to a sampling-based MC estimate, the paper also introduces an efficient one-shot approximation that leverages the asymptotic Gaussianity of the induced parameter posterior (i.e., when viewing it as a probabilistic model with a distribution over model parameters with deterministic LayerNorm, instead of stochastic LayerNorm with deterministic model parameters). Finally, the paper demonstrate the empirical effectiveness of the proposed MC-LayerNorm method on a set of standard calibration benchmarks (image classification on CIFAR-10-C, TinyImageNet-C, and ImageNet-C, Criteo Display Advertising Challenge) as compared to relevant baselines.

**Audience:**

Yes

**Claims And Evidence:**

Yes

**Requested Changes:**

N/A

**Strengths And Weaknesses:**

I think that this is a great and insightful paper that would be interesting to the TMLR audience. Its strengths include:
- The problem addressed (i.e., neural network calibration) is important and well-motivated.
- The manuscript is clearly written and easy to follow. The approach is placed in the context of relevant prior work.
- The proposed method is sound and intuitively sensible. Furthermore, being a drop-in replacement for a common neural network building block (LayerNorm), the approach is conceptually simple and straightforward to implement and can be used to obtain calibrated uncertainty estimates for a range of popular model architectures (especially models to which previous calibration methods were not applicable).
- The connection to approximate inference / Bayesian model averaging is derived clearly and makes the approach more principled.
- The empirical evaluation convincingly demonstrates the practical benefits of MC-LayerNorm in improving model calibration (especially on out-of-distribution inputs / under distribution shift) on relevant benchmarks as compared to previous methods. I especially commend the experiments on Vision Transformers due to their prevalence as one of the most important modern neural network architectures.

Overall, I thus recommend acceptance of the paper.

---

> ### Author Response · Authors · 2024-01-11
> **Thank you for your feedback!**
>
> We sincerely appreciate the reviewer's thoughtful feedback on our paper. We would like to thank the reviewer for acknowledging the clear writing, principled approach and our experiments on Vision Transformers.

---

### Review · Reviewer_ugqw · 2023-12-20

**Summary Of Contributions:**

This paper introduces MC Layer Normalization, a module enhancing neural networks' prediction uncertainty estimation. It serves as a replacement for Layer Normalization, targeting improved calibration in OOD scenarios. The approach is validated through various benchmarks and baselines.

**Audience:**

Yes

**Claims And Evidence:**

Yes

**Requested Changes:**

### Requested Changes
1. Provide detailed result tables.
2. Broaden comparative analysis.
3. Run with recent training routines.


### Final Comments
The paper offers an innovative approach for uncertainty estimation in neural networks, particularly for OOD scenarios. However, its impact seems incremental and empirical results are a bit weak. Enhancing training routines and providing detailed comparative analyses could strengthen the findings.

**Strengths And Weaknesses:**

**Strengths:**
1. **Innovative yet Simple Approach**: Combines successfully ideas from MC batch normalization and layer normalization.
2. **Practical Application**: Demonstrates some effectiveness across several tasks, particularly in OOD uncertainty estimation.
3. **Clear Introduction and Method Presentation**: Well-written with a lucid explanation of the method and related work.

**Weaknesses:**
1. **Limited Comparative Analysis**: Needs more comparison with recent methods like Function-Space Regularization (Rudner et al., 2023).
2. **Unclear Significance of Impact**: High error bars obscure the method's effectiveness; detailed result tables are recommended for clarity. Also, it can be useful to see the numbers and figures for the different corrupted  tasks and not only the average.
3. **Outdated Training Routines**: Lacks advanced training techniques like data augmentation and learning rate schedules.

---

> ### Author Response · Authors · 2024-01-11
> **Thank you for your feedback!**
>
> We sincerely appreciate the reviewer's thoughtful feedback on our paper. We would like to thank the reviewer for acknowledging the clear presentation and our methods innovative yet simple approach.
>
> Here below we address the raised weaknesses and detail how we tried to implement the requested changes in the revisions of our paper.
>
> ### Weakness 1
>
> > Limited Comparative Analysis
>
> We have run extensive additional experiments showing additional results for MC-BatchNorm, Masksemble and Temperature scaling for the two ConvNext settings (see updated Figure 1 and 2, as well as tables 1 and 2).
>
> We also considered Function-Space Regularization (Rudner et al., 2023), but we argue that this work is outside the scope of our analysis for a few reasons, the main one being that we are mostly interested in approaches that can be implemented without major overhaul of a given deep learning architecture and training pipeline. In fact, MC-LayerNorm can be directly dropped into a model by swapping out its LayerNorm modules, and everything else (loss, training schedule), remains the same. Moreover, the deployment workflow remains completely unaltered, as MC-LayerNorm at test-time is equivalent to LayerNorm in its one-shot approximation version. Function-Space Regularization on the other hand requires to modify the loss and the training procedure. We therefore argue that this method, as interesting as it is, is targeting a different set of use cases as ours. We however appreciate the Reviewer’s suggestion to consider this method and we cited this work in the Related Work section of the revised version of the paper where we discussed the points above, which we think help additionally clarify the scope and applicability space of our work.
>
> ### Weakness/Request 2
>
> > **Unclear Significance of Impact**: High error bars obscure the method's effectiveness; detailed result tables are recommended for clarity. Also, it can be useful to see the numbers and figures for the different corrupted tasks and not only the average.
>
> > Provide detailed result tables.
>
> As requested, we have included results in the form of tables in appendix A.2. This also gives us the chance of clarifying that the error bars in our plots are large due to the fact that they represent the distribution of performance across runs and across all the corruption levels. In contrast to the figures, the standard deviation in the tables is computed distinctively by first averaging across the multiple corruption levels, followed by calculating the standard deviation across training runs. For the figures, we followed the methodology of [R1] which calculates the error bars over the 15 corruptions applied to the original test set and over three training runs.
>
> The small standard deviations between models confirms that the large error bars are caused by differences between different corruptions and not from variances between models with the same parameters. This makes sense, as the corruptions range from light changes to jarring and with that model performance between different corruptions therefore varies widely.
>
> ### Weakness/Request 3
>
> > **Outdated Training Routines**: Lacks advanced training techniques like data augmentation and learning rate schedules.
>
> > Run with recent training routines.
>
> We consciously chose a minimal/basic training routine to decouple any influence of augmentations and learning rate schedule from the calibration results of the various methods.
>
> For training, we do use basic data augmentation for vision models: padding + random crop for CIFAR-10 and TinyImageNet while only taking a random crop for ImageNet. Additionally, we use a horizontal flip (p=0.5) augmentation for all vision datasets. No training augmentations are used for the Criteo models, in order to replicate the state-of-the-art setting for this benchmark. We have added these details in appendix A.4.

---

> > ### Author Response · Authors · 2024-01-11
> > **(Continued)**
> >
> > ### Final Comments
> >
> > >The paper offers an innovative approach for uncertainty estimation in neural networks, particularly for OOD scenarios. However, its impact seems incremental and empirical results are a bit weak. Enhancing training routines and providing detailed comparative analyses could strengthen the findings.
> >
> > As we argued in the paper, even though our empirical results are presented in opposition to related works such as MC-BatchNorm and MC-Dropout we don’t see them as in direct competition, but see them as complementary. In fact, our method can be seamlessly deployed in settings where MC-BatchNorm or MC-Dropout are not suitable, like for instance architectures such as transformer models or MaskNet where BatchNorm is not present but which on the other hand are endowed with LayerNorm modules. Trying to somehow force MC-BatchNorm in those models is known to lead to subpar performance, while MC-LayerNorm is a natural fit which requires minimal change in the training architecture and no change in the training routine. We therefore see our method as filling a gap in the current toolbox of uncertainty quantification techniques for deep learning, and therefore complementing and completing its current status, rather than incrementally adding to it. Moreover, since such models are becoming more and more prominent in modern deep learning, we believe that the gap that our MC-LayerNorm module is filling will become increasingly important.
> >
> > **References:**
> > [R1] - Prediction-Time BatchNorm https://arxiv.org/abs/2006.10963

---

### Review · Reviewer_RYYS · 2023-12-22

**Summary Of Contributions:**

Bayesian methods (e.g., Bayesian Deep Neural Networks) offers a principled way to estimate uncertainty, but due to its intractable computational cost, researchers have been proposed approximate Bayesian methods, e.g., Monte Carlo Dropout or Prediction-Time Batch Norm. However, these two methods are not technically applicable on some settings and lead to "suboptimal" performance. To address this issue, the paper proposes a new module, called MC Layer Normalization, which is a drop-in replacement for Layer Normalization to seamlessly add uncertainty calibration capability. The proposed module is theoretically motivated from an approximate Bayesian perspective, and empirically demonstrated its efficacy over two architectures and three datasets.

**Audience:**

Yes

**Broader Impact Concerns:**

no concerns.

**Claims And Evidence:**

No

**Requested Changes:**

* Provide intuition on why MC-LayerNorm should be better than other type of approximated Bayesian methods.
* Discuss whether approximate Bayesian methods are required for calibration and discuss pros and cons of approximate Bayesian methods and post-hoc calibration methods for calibrated uncertainty (in and out-of-distribution settings).
* Add temperature scaling results in Figure 1 as did for Figure 2.
* (if possible) Use the same set of baselines in Figure 1 and Figure 2.
* Discuss the reasons for the performance gap between T-Scaling and MC-LN+T-Scaling in Figure 2.
* Share the experiment setup for temperature scaling (e.g., loss, optimization iterations, calibration set size...).

**Additional editorial comments:**
* "conformal prediction methods Zhu & Rigotti (2021); Bates et al. (2021); Angelopoulos et al. (2022).": citations are wrong. Please acknowledge the original work, i.e., "Vladimir Vovk, Alex Gammerman, and Glenn Shafer. Algorithmic learning in a random world.
Springer Science & Business Media, 2005." or related papers.
* "(lower is better) quantifies the difference between the confidence of a model and its accuracy computed on bins of samples sorted by confidence Guo et al. (2017).": same here, Guo et al. (2017) adopts ECE from the earlier work.

**Strengths And Weaknesses:**

**Strengths**:

* The paper proposes a new module, MC-LayerNorm.
* The new module's asymptotic behavior is theoretically analyzed in Theorem 3.1.

**Weaknesses**:

* The motivation on why MC-LayerNorm could be better than other baselines is missing: to me MC-LayerNorm is still another type of approximated Bayesian methods, but it is unclear why it should be better than others. Can you provide more intuition?
* Related to the above concern, this paper's claim is contradictory to the existing observation in [R1] --- [R1] claims that "approximate Bayesian methods are not enough for calibration and we need the post-hoc calibration methods". This makes sense to me as the "approximation" in Bayesian methods manifests suboptimal results in calibration and we need the post-hoc calibration methods for the right uncertainty (in terms of the known perfect-calibration definition --- see Guo et al. (2017) for details). Can you discuss whether approximate Bayesian methods are required for calibration and discuss pros and cons of approximate Bayesian methods and post-hoc calibration methods for calibrated uncertainty (in and out-of-distribution settings)?
* An important baseline, temperature scaling, is missing in Figure 1. Can you add temperature scaling results in Figure 1 as did for Figure 2? It is even better if the same set of baselines are used in Figure 1 and Figure 2.
* It would be important to discuss on the performance gap between T-Scaling and MC-LN+T-Scaling in Figure 2, but it is missing: to me temperature scaling could be the most effective method for ECE as it directly minimizes the related quantity (i.e., I expect T-Scaling and MC-LN+T-Scaling ECE could be very similar at least in "Test"), but the results are counter-intuitive, which is interesting and important to highlight the proposed approach. Can you discuss the reasons for this performance difference? Regarding this, can you share the experiment setup for temperature scaling (e.g., loss, optimization iterations, calibration set size...).

[R1] https://proceedings.mlr.press/v80/kuleshov18a/kuleshov18a.pdf

---

> ### Author Response · Authors · 2024-01-11
> **Thank you for your feedback!**
>
> We sincerely appreciate the Reviewer's thoughtful feedback on our paper. We have revised the citations for conformal prediction methods and ECE as proposed in the editorial comment.
>
> Below we detail how our revisions implemented the changes requested by the Reviewer.
>
> ### Request 1
>
> > Request 1: Provide intuition on why MC-LayerNorm should be better than other type of approximated Bayesian methods.
>
> Our main motivation in proposing MC-LayerNorm is to provide a method that would be complementary to the already available toolbox of approximate Bayesian methods rather than in direct competition, particularly by being applicable in situations where previously proposed methods are not suitable. Concretely and as we mention also in the correspondence with other reviewers, modern state-of-the-art deep learning architectures often do not include any of the layers with a corresponding previously proposed uncertainty calibration counterpart. For instance, transformers and vision transformers do not feature BatchNorm, meaning that endowing them with MC-BatchNorm/Prediction-Time BatchNorm would require a relatively radical change in the architecture that will typically compromise on the performance and enforce a change in the training procedure (for instance by introducing constraint on the value of the mini-batch size hyperparameter to accommodate BatchNorm). Meanwhile, LayerNorm has become a pervasive normalization layer in modern deep learning architecture, making MC-LayerNorm a natural fit as a lightweight drop-in replacement for LayerNorm, meaning in particular that we don’t need to modify the training procedure and can even apply it to pretrained weights (which are then lightly fine-tuned).
>
> Beyond this main motivation, we do observe that MC-LayerNorm is often empirically better than other uncertainty calibrated normalization layers in settings where it makes sense to directly compare them. One of the reasons, is the fact that MC-LayerNorm does not need to rely on a data batch to be re-calibrated in an OOD setting (like for instance Prediction-Time BatchNorm does). As a consequence, it can immediately adapt to a situation where in-distribution data is suddenly followed by OOD data (what we called the ‘Mix’ case in the paper).
>
> ### Request 2
>
> > Related to the above concern, this paper's claim is contradictory to the existing observation in [R1] --- [R1] claims that "approximate Bayesian methods are not enough for calibration and we need the post-hoc calibration methods". This makes sense to me as the "approximation" in Bayesian methods manifests suboptimal results in calibration and we need the post-hoc calibration methods for the right uncertainty (in terms of the known perfect-calibration definition --- see Guo et al. (2017) for details). Can you discuss whether approximate Bayesian methods are required for calibration and discuss pros and cons of approximate Bayesian methods and post-hoc calibration methods for calibrated uncertainty (in and out-of-distribution settings)?
>
> > Request 2: Discuss whether approximate Bayesian methods are required for calibration and discuss pros and cons of approximate Bayesian methods and post-hoc calibration methods for calibrated uncertainty (in and out-of-distribution settings).
>
> We completely agree with the Reviewer and the provided claim from [R1] about the necessity of enhancing approximate Bayesian methods with post-hoc calibration. We indeed see these approaches as complementary and synergistic. For one, post-hoc calibration is not always possible as it requires a held-out in-distribution calibration dataset which might not always be available. In such a case, one necessarily has to rely on approaches like approximate Bayesian estimation methods. Secondly, post-hoc calibration only works if the base model is already at least partially calibrated, and the better calibrated the base model is, the more effective post-hoc calibration methods will be. This is another point emphasizing the importance of methods like ours, as they enable the applicability of post-hoc calibration.
>
> We thank the Reviewer for raising these points of discussion which we addressed by clarifying the above arguments in the text.
>
> We also emphasized that this viewpoint along with the claim from R1 is consistent with the results from Figure 1 and 2 which show that the combination of MC-LN and Temperature Scaling results in the best results, which supports their synergistic relationship.
>
> ### Request 3
>
> > Request 3: Add temperature scaling results in Figure 1 as did for Figure 2. (if possible) Use the same set of baselines in Figure 1 and Figure 2.
>
> We have run extensive additional experiments showing results for MC-BatchNorm, Masksemble and Temperature scaling for the two ConvNext settings (see updated Figure 1 and 2, as well as tables 1 and 2).

---

> > ### Author Response · Authors · 2024-01-11
> > **(Continued)**
> >
> > ### Request 4
> >
> > > Request 4: Discuss the reasons for the performance gap between T-Scaling and MC-LN+T-Scaling in Figure 2.
> >
> > We agree that this is an interesting question and have added a paragraph in section 4 discussing this observation. We believe that there can be a number of reasons for this performance gap. To echo a bit the previous point on the synergy between pre- and post-hoc calibration methods, Temperature Scaling is most effective when the base model is already at least partially calibrated, at least to the extent that the miscalibration pattern is consistent across the range of logits. If the base model's miscalibration varies widely across classes or confidence levels, temperature scaling might not be sufficient. This is where we believe that approximate Bayesian models can certainly be crucial to enable further calibration through post-hoc methods.
> >
> > This gap in performance is also consistent with experiments in [R2] where pattern between Prediction-BatchNorm and Prediction-BatchNorm + Temperature Scaling.
> >
> > ### Request 5 (Request 4 in original Review)
> >
> > > Request 5: Share the experiment setup for temperature scaling (e.g., loss, optimization iterations, calibration set size...).
> >
> > We have added section in Appendix 4 outlining the parameters used for Temperature Scaling which we are inspired by [R3]:  NLL loss, opt=LBFGS, lr=0.01, max_iter=50
> >
> > ### Additional editorial comments
> >
> > We thank the Reviewer for helping rectifying our citation list. The papers we cited where meant to provide some works on conformal prediction methods specifically applied to deep learning. We agree with the Reviewer that this is however an idiosyncratic and partial view of the literature and we therefore added citations to the provided original work by Vovk and Gammerman.
> >
> > As already indicated, we also added the provided reference on ECE, Guo et al. (2017), Naeini et al. (2015)
> >
> > **References**
> >
> > [R1] - [https://proceedings.mlr.press/v80/kuleshov18a/kuleshov18a.pdf](https://proceedings.mlr.press/v80/kuleshov18a/kuleshov18a.pdf)
> >
> > [R2] - Prediction-Time Batch Norm https://arxiv.org/abs/2006.10963
> >
> > [R3] - [https://github.com/gpleiss/temperature_scaling](https://github.com/gpleiss/temperature_scaling) (Implementation of Gao et. al)

---

### Review · Reviewer_NxVm · 2023-12-24

**Summary Of Contributions:**

Summary:

This work proposes a new normalization module, the MC Layer Normalization module, with the goal of calibrating the uncertainty estimation in deep learning models. Differing from traditional Layer Normalization, MC-LayerNorm samples randomly from the units in a given layer and computes the normalization statistics over the sampled units. This allows for a more effective estimation of prediction uncertainty, particularly in scenarios of distribution shift. The module's theoretical design relies on combining Bayesian modeling and Monte Carlo integration. To show empirical evidence of the effectiveness of the proposed layer, this work tests the MC-LayerNorm in different settings where out-of-distribution examples are present, challenging the confidence of the model over these examples.

**Audience:**

Yes

**Claims And Evidence:**

No

**Requested Changes:**

See questions listed as part of the weaknesses.

**Strengths And Weaknesses:**

Strengths:

- Estimating/calibrating the uncertainty/confidence is a major challenge in neural network predictions, and this paper addresses it with an intuitive and simple-to-implement  approach.
- Overall, the paper is well-written and clear to follow.
- The related work is well written and it highlights both existing work in the scope of the paper and the existing parallel works to estimating uncertainty of neural networks.
- This work shows empirically a relative enhancement in the performance of the proposed method in different settings, challenging existing the baselines. However, the major advantage of this method is that it can be applied to several settings unlike the shown baselines.

Weakness:
- This paper claims that comparing to Prediction-Time Batch Normalization is sufficient, however, a work such as (Monte Carlo Batch Normalization (Teye et al. (2018)) or the Temperature scaling should be present in the experimental section in all settings. Alternatively, a thorough discussion of why MC-LayerNorm is better suited for uncertainty estimation should be provided.
- In settings 2 and 3 of the experimental section, the paper does not clarify well why "MC-LayerNorm is the only norm layer based uncertainty calibration method that is applicable to this architecture”,  or why it is only compared to “the baseline of the traditional LayerNorm” in the Criteo challenge?
- What is the effect of the relative $S_l$ set size to $N_l$ on the final performance on both In-distribution (Test) and Out-of-distribution (OOD) test when it is closer to 0? When the ratio is larger than 70%, is there an explanation why the performance is very stable? Also why is this ablation not present on the Criteo dataset experiment?
- Temperature scaling showed better performance on Vision Transformer, how about ConvNext? The higher performance of the  combination of MC-LayerNorm and post-hoc Temperature scaling seems to be caused by the temperature scaling itself.

Minor comment:
Citations should be inside parentheses when they are not to be read as part of the text.

---

> ### Author Response · Authors · 2024-01-11
> **Thank you for your feedback!**
>
> We sincerely appreciate the Reviewer's thoughtful feedback on our paper. We would like to thank the Reviewer for acknowledging the clear writing, the related works section and the intuitiveness of our approach.
>
> Here below we address the specific comments provided by the Reviewer:
>
> >Citations should be inside parentheses when they are not to be read as part of the text.
>
> Thank you, we have fixed this citation issue throughout the paper.
>
> ### Weakness 1
>
> > This paper claims that comparing to Prediction-Time Batch Normalization is sufficient, however, a work such as (Monte Carlo Batch Normalization (Teye et al. (2018)) or the Temperature scaling should be present in the experimental section in all settings. Alternatively, a thorough discussion of why MC-LayerNorm is better suited for uncertainty estimation should be provided.
>
> Following the Reviewer’s comment, we have conducted extensive additional experiments, presenting additional results for MC-BatchNorm, Masksemble, and Temperature Scaling in the two ConvNext settings (refer to the updated Figure 1 and 2, as well as the Tables in the appendix).We also detailed more clearly in the text what we mean when we say that MC-LayerNorm is “better suited" for uncertainty estimation. In short, MC-LayerNorm is a better fit for architectures that already have a LayerNorm layer and for which Dropout and BatchNorm (and by extent MC-Dropout and MC-BatchNorm/Prediction-Time BatchNorm) are not an option. In these cases, MC-LayerNorm is a natural and straight-forward method that can be dropped-in with basically no modification of the architecture and training procedure. ViT’s are such an example, as Layer Normalization is typically preferred over BatchNorm in that setting. A more comprehensive explanation follows in the response to weakness 2.
>
> ### Weakness 2
>
> > In settings 2 and 3 of the experimental section, the paper does not clarify well why "MC-LayerNorm is the only norm layer based uncertainty calibration method that is applicable to this architecture”, or why it is only compared to “the baseline of the traditional LayerNorm” in the Criteo challenge?
>
> Thank you for indicating this point in need of clarification. In the text we added additional information in section 4 providing the needed clarifying details.
>
> To summarize, as we alluded to in response to the previous comment, with “applicable to architecture”, we mean that the state-of-the-art implementation (and thus also pre-trained models) of those architectures do not include any of the layers with a corresponding previously proposed uncertainty calibration counterpart. In particular, the SOTA architecture for the Criteo challenge does not include a BatchNormalization layer, meaning that it cannot be naturally extended with a MC-BatchNorm/Predition-Time BatchNorm layer without modifying the training procedure (specifically, to take into account the effect of mini-batch size on the training with BatchNorm), and without incurring a loss in performance (as SOTA is empirically achieved without a BatchNorm layer). The SOTA architecture for Criteo however includes LayerNorm modules, meaning that we can straightforwardly replace them by dropping in MC-LayerNorm without changing anything else in the architecture or training, and even in the use of the architecture at test-time if we use the one-shot test-time version of MC-LayerNorm.
>
> The Criteo challenge is therefore meant as a use case to emphasize the advantages of our approach also beyond vision tasks, in particular its flexibility as a drop-in replacement layer for LayerNorm (a normalization layer which is almost ubiquitous in modern deep learning), its ease of use since it does not require any other modifications to the models architecture and can be applied to pre-trained models with only minimal additional fine-tuning, and its broad applicability which makes it the ideal complementary method to the previously proposed methods such as MC-Dropout and MC-BatchNorm/Prediction-Time BatchNorm when these are not a natural fit to the architecture under consideration.

---

> > ### Author Response · Authors · 2024-01-11
> > **(Continued)**
> >
> > ### Weakness 3
> >
> > > What is the effect of the relative $\mathcal{S}_l$ set size to $N_l$ on the final performance on both In-distribution (Test) and Out-of-distribution (OOD) test when it is closer to 0? When the ratio is larger than 70%, is there an explanation why the performance is very stable? Also why is this ablation not present on the Criteo dataset experiment?
> >
> > The hyperparameter $f=\mathcal{S}_l/{N_l}$ indicates the fraction of neurons over which we compute the normalization constants. The limit where f goes to zero therefore corresponds to not using any units for normalization, i.e. simply completely removing the normalization layer. That is a well-defined limit which however empirically results in architectures that do not train well, since they suffer of the covariate shift problem that is solved by normalization layers. The opposite limit about which the Reviewer is asking, f going to 1, corresponds to using all features for normalization, which simply converges to LayerNorm. This also clarifies why performance is stable in this limit: we simply converge towards the vanilla LayerNorm case, which is well-defined and robust. As for the Criteo use case, we verified and clarified in the text that the reported performance is robust within the interval f=0.6-0.8, with a gradual progressive degradation as f is varied above or below this range.
> >
> > ### Weakness 4
> >
> > > Temperature scaling showed better performance on Vision Transformer, how about ConvNext? The higher performance of the combination of MC-LayerNorm and post-hoc Temperature scaling seems to be caused by the temperature scaling itself.
> >
> > As mentioned above, we have run additional experiments to address the Reviewer’s comment (including Temperature scaling for ConvNext settings) and added an additional discussion of this issue in section 4.
> >
> > In regard to the effect of temperature scaling, we believe that it is important to distinguish between methods that improve the intrinsic calibration of the model, and methods that are applied post-hoc and require a held-out calibration dataset (like temperature scaling). First of all, these two classes of methods naturally complement each other: as our experiments show MC-LayerNorm works well in tandem with temperature scaling, resulting in a better overall calibration. This is expected, but not granted as a positive contribution from temperature scaling is predicated on a base model whose calibration is consistent across confidence levels and classes, a premise that our results verify. What’s more, even though the Reviewer is correct that temperature scaling alone does indeed show a strong effect in several cases, in most plots the combined effect of temperature and MC-LayerNorm is noticeably more robust, indicating a practically interesting synergistic effects between the two methods. Finally, post-hoc calibration methods like temperature scaling are not always available, since they require held-out in-distribution calibration datasets, in absence of which we can only rely on intrinsic calibration methods like the one we propose.

---

### Decision · Action_Editor_FHHa · 2024-02-06

**Recommendation:** Accept as is

**Comment:**

This paper presents a drop-in replacement for layer normalization blocks, referred to as "MC layer normalization" so that the neural networks are endowed with uncertainty quantification capabilities. Transformers contain layer normalization blocks, so that the proposed MC layer normalization could be very useful for many practitioners. The proposed method is not the best UQ technique, but adds one easy-to-use entry to the available toolbox of approximate Bayesian methods. One of shortcomings here is the necessity of enhancing the MC layer normalization  with post-hoc calibration, since it not always possible to have a sufficient amount of held-out in-distribution calibration dataset. Despite this limitation, the method here is complementary and synergistic. Thus, the paper contains sufficient interesting contributions.

**Audience:**

This paper provides a method that would be complementary to the already available toolbox of approximate Bayesian methods for uncertainty quantification. A drop-in replacement for layer normalization blocks, proposed in this paper, will be very useful for many TMLR's audience, since it endows a neural network (in particular transformers) with uncertainty quantification capabilities.

**Claims And Evidence:**

The paper makes a valuable contribution by introducing MC layer normalization to address uncertainty calibration in deep learning models. All of reviewers agree that the technique introduced here is simple but valuable and is also supported by Theorem 3.1. In order to accommodate the feedback by reviewers, the authors made efforts to add more experiments, clarify the concerns. Without any doubt, the claims are sound and convincing.